# ReNF: Rethinking the Design of Neural Long-Term Time Series Forecasters

Yihang Lu [1 2]   Xianwei Meng [1 3]   Enhong Chen [2]

## Abstract

Neural Forecasters (NFs) have become a cornerstone of Long-term Time Series Forecasting (LTSF). However, recent progress has been hampered by an overemphasis on architectural complexity at the expense of fundamental forecasting structures. In this work, we revisit principled designs of LTSF. We begin by formulating a Variance Reduction Hypothesis (VRH), positing that generating and combining multiple forecasts is essential to reducing the inherent uncertainty of NFs. Guided by this, we propose Boosted Direct Output (BDO), a streamlined paradigm that synergistically hybridizes the causal structure of Auto-Regressive (AR) with the stability of Direct Output (DO), while implicitly realizing the principle of forecast combination within a single network. Furthermore, we mitigate a critical validation-test generalization gap by employing parameter smoothing to stabilize optimization. Extensive experiments demonstrate that these trivial yet principled improvements enable a direct temporal MLP to outperform recent, complex state-of-the-art models in nearly all benchmarks, without relying on intricate inductive biases. Finally, we empirically verify our hypothesis, establishing a dynamic performance bound that highlights promising directions for future research. The code is publicly available at: https://github.com/Luoauoa/ReNF.

## 1. Introduction

The progression of any real-world event is a unique, non-repeatable process, often governed by chaotic dynamics that we cannot perfectly measure (Robertson, 1929) or describe. This implies that any observed time series is a single stochastic realization of a complex underlying system, inevitably corrupted by measurement errors. Consequently, a fundamental open question persists: how can we best estimate long-term future states using a data-driven model that relies solely on a single observed historical sequence?

Deep Neural Networks (DNNs) have recently received considerable critical attention in Long-Term Time Series Forecasting (LTSF) (Kim et al., 2025). Their capacity to model high-dimensional, non-linear dependencies makes the Neural Forecaster (NF) a promising tool for capturing complex temporal dynamics (Laiz et al., 2024). However, the literature reveals several critical bottlenecks that have to be addressed to unlock the full potential of DNNs in this domain.

A confounding paradox in recent research is that advanced architectures, such as Transformer-based models (Nie et al., 2023), and simple linear models (Zeng et al., 2023) are reported to achieve state-of-the-art performance interchangeably, despite vast differences in complexity. This is partly because some datasets favor parsimonious models (Deng et al., 2024), while others possess sufficient volume to fit complex architectures. Yet, a deeper issue may be the insufficient exploration of the networks' intrinsic capabilities. As indicated by (Lu et al., 2025), many complex NFs contain redundant components that are activated selectively based on data properties, leaving their full potential untapped without extensive tuning.

Furthermore, the field has shifted towards designing specialized modules for specific properties, such as multi-scale modeling (Wang et al., 2024b) and non-stationarity adaptation (Liu et al., 2022). However, progress has become erratic, as these architectural additions often yield subtle gains while overlooking fundamental principles. A primary and more general path to advancement may lie in fundamentally improving the training stability and generalization capabilities of NFs themselves.

Notably, training on time series often suffers from significant instability. The non-stationary nature of real-world time series, combined with chronological data splitting, leads to both internal (batch-to-batch) and external (train/validation/test) distribution mismatches. This causes the optimization path to be heterogeneous, rendering the learning process for NFs ineffective. Specifically, internal data redundancy can cause the model to overfit spurious patterns (Liu et al., 2025a), while external distribution shift

---

[1]HFIPS, Chinese Academy of Sciences [2]University of Science and Technology of China [3]Hefei University of Technology. Correspondence to: Xianwei Meng <mengxw@iim.ac.cn>.

*Proceedings of the 43rd International Conference on Machine Learning*, Seoul, South Korea. PMLR 306, 2026. Copyright 2026 by the author(s).

creates inconsistencies across training phases. Although normalization techniques (Kim et al., 2021) address train-test shifts, a critical discrepancy remains between validation and testing performance, which insidiously prevents NFs from demonstrating their true capabilities, and even disturbs further investigations like ablation studies (see Sec. 2.5). To mitigate this problem, we advocate for the employment of weight averaging techniques (Ruppert, 1988; Izmailov et al., 2018; Brotons et al., 2024) within the practical training pipeline of neural forecasters. Specifically, we demonstrate that applying an efficient Exponential Moving Average (EMA) to track model parameters smooths the optimization trajectory, ultimately leading to more robust and generalized forecasters.

Beyond model architecture, we identify a structural limitation in the dominant Direct Output (DO) framework: it fails to fully utilize available supervision. In a standard DO setup, an NF is trained as a single-input single-output system to predict the entire future horizon in a single forward pass, meaning the whole label information is leveraged only once per optimization step. This may tend to encourage the model to learn a monolithic structural representation that maps history directly to the future, without explicitly modeling the sequential dependencies within the forecast itself. In other words, the model seems to fit regressions at each target time step independently while ignoring the inherent order of its outputs. We contend that this approach hinders the model from developing a more granular, causal understanding of the future, thereby limiting its full potential.

To address these fundamental issues, this paper proposes some principled improvements that help establish a reliable and high-performing NF. Our contributions are summarized as follows:

- We redesign the forecasting structure with a novel, streamlined stacking paradigm that synergistically combines the causal structure of Auto-Regressive (AR) methods with the stability of Direct Output (DO).

- We identify the validation-test generalization gap as a critical bottleneck in evaluating Neural Forecasters. To bridge this gap, we adopt Exponential Moving Average (EMA), demonstrating that this standard technique is particularly effective in the LTSF domain for stabilizing optimization and preventing spurious overfitting.

- We conduct extensive experiments to evaluate the effectiveness of our proposals and demonstrate that a pure MLP-based forecaster, when trained with our paradigm, can still outperform recent state-of-the-art models on nearly all standard LTSF benchmarks.

## 2. Method

In this section, we lay out the methodological foundations of our work. To build intuition, we highlight the key effects of each consideration with empirical demonstrations using a toy MLP on selected datasets.

**Problem Statement.** We define a Neural Forecasting Machine (NFM) as a random function $\Phi_\theta(X, \gamma) \to Y$, where $\theta$ denotes learnable parameters and $\gamma$ represents the model's stochastic state. Given a dataset of finite realizations drawn from an underlying process, partitioned into history $X_h \in \mathbb{R}^{T_x \times D}$ and future $X_f \in \mathbb{R}^{T_y \times D}$, where $T$ and $D$ denote the temporal and variate dimensions, respectively. The forecasting task aims to learn an NFM such that the generated prediction $\hat{Y}_f = \Phi_\theta(X_h, \gamma)$ minimizes the error metric between $\hat{Y}_f$ and $X_f$.

### 2.1. Theoretical Motivation for Multiple Forecasts

We begin by introducing a core hypothesis that provides the primary motivation for building a multi-forecast system, which allows the trained NFM to act as a generator capable of producing *multiple outputs* for a fixed input, conditioned on varying internal states or configurations. To theoretically assess the capability of such a forecaster, we estimate the point-wise error between the generated predictions $\hat{Y}_f$ and the expected future $X_f$ as follows.

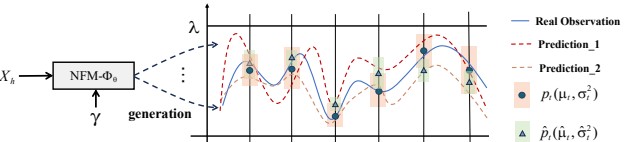

*Figure 1.* A single trained NFM $\Phi_\theta$ can generate multiple forecasts from a fixed input $X_h$ under various states $\gamma$ and $\theta$. These forecasts are expected to follow the empirical process $(\hat{p}_t, \ t = 1, 2, \cdots)$ approximated by the NFM with the observed data. The model-estimated process is distinguished from the real data distribution $(p_t, \ t = 1, 2, \cdots)$ by a bias $b_t$.

**Hypothesis 2.1** (Variance Reduction Hypothesis (VRH)). Given a NFM $\Phi_\theta$ and an observed time series $X_h = (x_1, x_2, \cdots, x_h)$ where each element $x_t$ is drawn from the true distribution $p_t(\mu_t, \sigma_t^2)$ with mean $\mu$ and standard deviation $\sigma_t$. We hypothesize that the expected error of the ensemble result derived from candidate forecasts generated by $\Phi_\theta$ with each element $\hat{y}_t$ drawn from the estimated future distribution $\hat{p}(\hat{\mu}_t, \hat{\sigma}_t^2)$ is dominated by a upper bound, in which an inherent predictive variance exists and scales inversely with the number of forecasts,

$$( \underbrace{\sigma}_{\text{data noise}} + \underbrace{b}_{\text{predictive bias}} + \underbrace{\lambda/\sqrt{c}}_{\text{predictive variance}} ) * L \quad (1)$$

where $\sigma$ and $b$ are the bounds on data noise and expected model bias, respectively. $\lambda$ can be the bound of the second

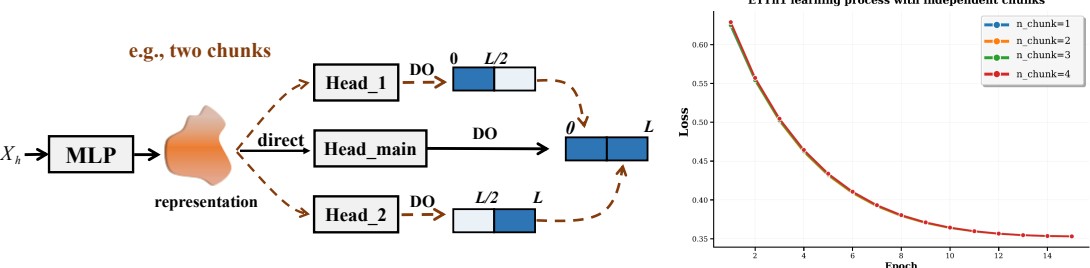

*Figure 2.* We compare the forecast by applying independent linear heads to predict several non-overlapped chunks of the horizon using a shared representation, with a main head for the whole horizon using the same representation. The right figure shows that the training loss curves in different settings are nearly identical under the DO strategy, indicating a lack of temporal dependencies.

moment of the forecasts. $L$ and $c$ denote the prediction horizon and number of forecasts, respectively.

*Remark* 2.2. *This bound relies on independence assumptions, which simplifies the dependencies inherent in our empirical design. Nonetheless, the analysis provides a qualitative guide, **exposing the problem of inherent predictive variance and motivating the shift from single-shot to multi-output forecasters.***

The proof is provided in Appendix B.1. To interpret this bound value, we decompose the error into three distinct components. First, $\sigma$ represents the intrinsic data noise, which is irreducible by the model. Second, the predictive bias reflects the approximation gap between the learned and true distributions; while this can be reduced by enhancing model capacity, it has been the primary focus of prior research. Crucially, we argue that the final term, the predictive variance, is largely overlooked in the design of deterministic neural forecasters. By default, these models yield a single output ($c = 1$), leaving this error component from the inherent uncertainty of a neural network unresolved, and it would be increasingly significant as the horizon $N$ prolongs.

**Post combination of multiple forecasts.** Given a set of candidate outputs from an NFM, there exist aggregation strategies to synthesize a single, integrated forecast that is more accurate than any individual candidate (Clemen, 1989). We represent this aggregation as a functional $g_c$ that produces the post-combined forecast:

$$\hat{Y}_{pc} = g_c(\{\hat{Y}_f^{(i)}\}_{i=1}^c) \qquad (2)$$

In Sec. 3.3, we empirically demonstrate that the performance of an 'oracle' combination on real-world datasets is consistent with our hypothesis. However, constructing an optimal combination in practice is non-trivial (Clemen, 1989) since intricate combination methods practically underperform the simple average (Stock & Watson, 2004). Consequently, a primary objective of our proposed framework is to bypass explicit combination design and instead enable the forecasting paradigm to *implicitly* learn itself how to leverage the information contained within these multiple forecasts, while maintaining the methodological parsimony.

## 2.2. Novel Forecasting Paradigm

The current frameworks in LTSF involve two main strategies: 1-step Auto Regressive (AR(1)) and Direct Output (DO). While AR models perform consistently with recurrent state space models like RNNs (Siami-Namini et al., 2019), they are known to suffer from significant error accumulation in long-horizon forecasting and are empirically evidenced to largely fall behind DO in LTSF (Zhou et al., 2021), making a strong reason for the dominant position of DO in LTSF. However, the shortcomings of the DO approach itself have been largely overlooked. We argue that despite its simplicity, the most significant weakness of DO, particularly when compared to AR, is its lack of inherent causality in the structure of the predictive representations.

To illustrate this, we conduct a simple experiment in Fig.2: we consecutively split a forecast of length $L$ into $m$ non-overlapped chunks with length $L/m$, and each is predicted by an independent linear head from a shared representation. The final forecast is a concatenation of these segments. Empirically, it turns out that the performance of this multi-headed model is nearly identical to a standard DO model with a single head predicting the full horizon. This result suggests that the NF is almost unaware of the temporal relationships within the future sequence and makes the prediction without considering the interval between the input history and the output future. As a result, it learns to forecast each segment independently, which seems counter-intuitive.

According to this observation, and combining our heuristic of multiple forecasts from VRH, we propose a new forecasting paradigm as follows,

**Definition 2.3** (Boosted Direct Output (BDO)). Given a history time series $X = \{x_1, x_2, \cdots, x_p\}$ (abbr. $X\{1 : p\}$) with length $h$, the NF recursively generates the forecast $\hat{Y}$ of the future object $Y = \{y_1, y_2, \cdots, y_L\}$ over $N$-steps. Let $\hat{Y}\{1 : h_{k-1}\}$ be the forecast at step $k - 1$, then the forecast at step $n$ is generated by:

$$\hat{Y}\{1 : h_k\} = \text{NF}([X, \hat{Y}\{1 : h_{k-1}\}]), \ k = 1, 2, \cdots, N; \qquad (3)$$

where $h_0 = 0$ and $h_N = L$. $[\cdot, \cdot]$ indicates the concatenation along the temporal dimension. In our implementation, we evenly split the forecasting length into $N$ segments for convenience, i.e., in Eq. 3, $h_N = h * N$, where $N$ should be a factor of $L$.

*Remark 2.4. BDO generalizes DO but is distinct from AR. Unlike existing AR paradigms (Salinas et al., 2020) which feed predictions back to yield the next step consecutively,* **BDO regenerates the horizon from the start point at each iteration.** *Thus, every intermediate output is a complete sub-forecast rather than a dependent, successive fragment of a single series.*

Intuitively, BDO recursively generates forecasts for progressively longer horizons, reusing predictions from previous stages. This incorporates an AR-like causal structure into the forecasting process, while retaining the patch-wise output characteristic of DO that mitigates significant error accumulation. A balance between these two properties can be achieved by properly setting the number of recursive stages, $N$ (see Appendix B.2 for theoretical analysis). Furthermore, since short-term forecasting is generally an easier task than long-term forecasting, BDO effectively creates a learning curriculum and tends to build representations with a hierarchical structure (see Fig.11). We can reasonably expect this paradigm to outperform both pure AR and pure DO strategies, especially in the challenging LTSF setting.

### 2.3. Model Architecture

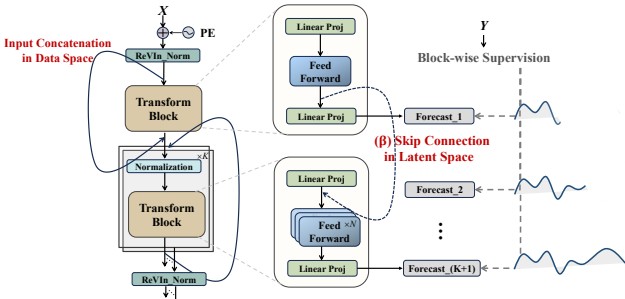

*Figure 3.* Model structure of ReNF. "PE" denotes optional temporal embedding, and the $\beta$ version adds additional skip-connections for deeper representations.

We construct our NFs using only MLP and linear layers for: 1) As foundational deep learning modules, improvements demonstrated on them are broadly applicable and convincing. 2) Recent work has shown that MLP-based architectures perform well for a wide range of forecasting tasks (Ekambaram et al., 2023).

Fundamentally, we build two NF variants, ReNF-$\alpha$ and ReNF-$\beta$, which differ in their degree of non-linearity for handling datasets of varying complexity. The overall architecture is shown in Fig.3, which is characterized by two main principles.

**Block-wise Supervision.** To adopt the BDO strategy, we stack multiple blocks, and each consists of a linear layer for representation projection, followed by an MLP for nonlinear transformation. Particularly, the end of each block is equipped with a dedicated linear head that maps the internal representation back to the data space, steering it to function as a **sub-forecaster** for horizon $h_k$.

**Implicit Forecast Combination.** In the recursive BDO process, each sub-forecast is concatenated with the original input to form the new input for the next block. This operation is trivial yet crucial in our design as it allows an implicit generalized forecast combination as follows,

$$\hat{Y}\{1 : h_k\} = \text{NF}([X, \hat{Y}\{1 : h_{k-1}\}])$$

$$= \left[ \underbrace{X}_{\text{Input}} \quad \underbrace{\hat{Y}\{1 : h_{k-1}\}}_{\text{Last Forecast}} \right] \cdot \left[ \frac{f}{\varphi} \right]$$

$$= f(X) + \varphi(\hat{Y}\{1 : h_{k-1}\})$$

where we can intuitively interpret the $k$-th MLP block as a composed operator with a predictor $f : \mathbb{R}^h \to \mathbb{R}^{h_k}$ and an adaptor $\varphi : \mathbb{R}^{h_{k-1}} \to \mathbb{R}^{h_k}$. In this sense, if we densely connect all outputs, every intermediate stage of the NF will recursively learn the implicit combination and finally yield a complex composed forecast.

We employ RevIN (Kim et al., 2021) as the pre-normalization for the initial input data to reduce the distribution discrepancy between the training and evaluation phases. For consistency, we also apply pre-normalizations to the input of each sub-forecaster. Additionally, we apply dropout before the initial linear projection to prevent the model from being overly dependent on the history observation and the concatenated information in the data space. To allow for representation flow with diverse depths, the ReNF-$\beta$ variant also incorporates skip-connections between the representation spaces.

The general equation for the transformation process within each sub-forecaster can be written as follows,

$$\text{ReNF\_Block}(X) = \text{Proj}(\text{Transform}(\text{Proj}(\text{Norm}(\text{Drop}(X)))))$$

where $X$ denotes the input series, and all the projections are performed on the temporal dimension.

### 2.4. Learning Objective

To train a deterministic NF, we adopt the hybrid loss function as used in (Liu et al., 2025a), which is a convex combination of the Mean Absolute Error (MAE) in both the time and frequency domains. The frequency-domain component has been shown to be effective at reducing spurious autocorrelations in the labels (Wang et al., 2024a).

Our BDO framework uniquely generates multiple outputs of varying lengths, enabling us to apply this loss at each

forecasting stage. This hierarchical supervision allows us to fully leverage the label information at multiple scales, encouraging the model to build causally structured and homogeneous representations rather than the disconnected ones typical of standard DO forecasts.

To ensure a stable foundation as well as the diversity of different forecasts, we place heavier weights on the losses of earlier short-term sub-forecasts. The complete loss function is expressed as follows:

$$loss = \sum_{n=1}^{N} (\gamma/n) * (\alpha * \|\hat{Y}_f^{(n)} - X_f^{(n)}\|_1 \quad (4)$$
$$+ (1-\alpha) * \|\text{Freq}(\hat{Y}_f^{(n)}) - \text{Freq}(X_f^{(n)})\|_1)$$

where $\| \cdot \|_1$ denotes the $l_1$ norm, $\gamma$ and $\alpha$ are predefined hyperparameters. $\text{Freq}(\cdot)$ represents the discrete Fourier transform and $\hat{Y}_f^{(n)}$ denotes the $n$-th sub-forecast corresponding to the $n$-th block of the model. A more in-depth analysis of this loss function is provided in Appendix C.

## 2.5. Smoothing the Learning Trajectory for Time Series

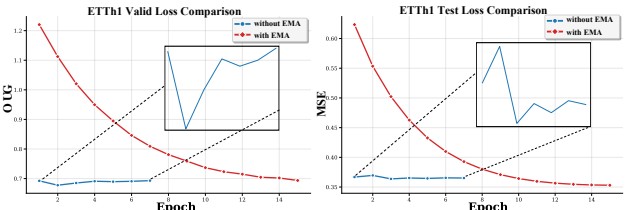

*Figure 4.* Variation of valid and test loss with and without applying EMA weight averaging. The valid loss and test loss manifest significant inconsistency during the learning process of NFs without smoothing, biasing the underlying evaluations of NFs.

On many time series datasets, particularly those with smaller volumes, the validation and test losses exhibit significantly different and, at times, conflicting dynamics during training. This issue invalidates the early stopping criterion: learning steps become ineffective, and suboptimal models are saved, preventing a true assessment of a model's capabilities. This is especially problematic when comparing models of varying complexities, which naturally have different optimization trajectories.

To mitigate these unexpected effects, we propose smoothing the training trajectory by employing an averaged model for evaluation. We achieve this efficiently using an Exponential Moving Average (EMA) to track the parameters of the online model (Izmailov et al., 2018). While this has been a standard technique in self-supervised learning (He et al., 2020) and generative modeling (Song et al., 2020), its role in mitigating the specific distribution shifts found in LTSF has been largely underexplored. We empirically establish it as a useful tool for robust performance. Specifically, let $\theta$ be the

parameters of the online model being trained, and $\theta'$ be the parameters of the shadow model. Then the shadow model's parameters are upgraded as follows at each iteration.

$$\theta'_{\text{new}} = \alpha * \theta'_{\text{prev}} + (1 - \alpha) * \theta_{\text{current}} \quad (5)$$

where $\alpha$ is the EMA decay rate. This shadow model is then used for all evaluations and early stopping. As shown in Fig. 4, this technique effectively smooths the learning curves and mitigates the inconsistency between validation and test performance. By providing a more stable and reliable training signal, EMA prolongs the effective learning period and may allow the model to converge to better local minima.

## 3. Experiment

**Baselines and Datasets.** Our empirical evaluation is designed to first establish the core strengths of our method in its intended domain and then to verify its generality across a wider range of forecasting tasks. Accordingly, the main text concentrates on established multivariate long-term forecasting benchmarks (ETTh, Electricity, Weather, etc.), where the hierarchical, variance-mitigating structure of BDO is particularly effective. To validate the versatility of our approach, we then present a comprehensive suite of supplementary experiments in the appendix. These evaluations test our proposals on distinct domains, including diverse short-term forecasting datasets from Kaggle (Yue et al., 2025), spatio-temporal traffic data (PEMS), and classical univariate series from M4 competition (Wang et al., 2024b). A complete description of these datasets can be found in Appendix A.

For the multivariate long-term forecasting, we compare our model with a suite of recent state-of-the-art methods, including TimeBridge (Liu et al., 2025a), DUET (Qiu et al., 2025), TimeDistill (Ni et al., 2026), Timer-XL (Liu et al., 2025b), iTransformer (Liu et al., 2024), TimeMixer (Wang et al., 2024b), PatchTST (Nie et al., 2023), Crossformer (Zhang & Yan, 2023), and Dlinear (Zeng et al., 2023).

**Setups.** All experiments were conducted on a single NVIDIA 4090 GPU with 24GB of memory, using the Adam optimizer (Kingma & Ba, 2014) and a fixed random seed of 2021 for reproducibility. Results for all baseline models were reproduced using their official source code and optimal configurations. For our model, ReNF, we searched for the optimal learning rate in the range from 0.0001 to 0.005, the EMA decay rates in the range from 0.99 to 0.999, and the number of layers from 2 to 8. We apply ReNF-$\alpha$ to ETTh1 and ETTh2 datasets, and ReNF-$\beta$ to others. The look-back window of all models is searched over $\{336, 512, 720\}$ for the best long-term forecasting performance following the standard benchmark configuration (Qiu et al., 2025).

*Table 1.* Results of long-term forecasting of hyperparameter searching. All results are averaged across four different prediction lengths: {96, 192, 336, 720}. The best and second-best results are highlighted in **red** and blue, respectively. Full results are listed in Appendix D.

| Models | ReNF ours | | TimeBridge ((2025a)) | | DUET (2025) | | TimeDistill (2026) | | Timer-XL (2025b) | | iTransformer (2024) | | TimeMixer (2024b) | | PatchTST (2023) | | Crossformer (2023) | | DLinear (2023) | |
|---|---|---|---|---|---|---|---|---|---|---|---|---|---|---|---|---|---|---|---|---|
| Metric | MSE | MAE | MSE | MAE | MSE | MAE | MSE | MAE | MSE | MAE | MSE | MAE | MSE | MAE | MSE | MAE | MSE | MAE | MSE | MAE |
| Weather | **0.214** | **0.247** | 0.220 | 0.250 | 0.219 | 0.253 | 0.221 | 0.269 | 0.240 | 0.273 | 0.232 | 0.269 | 0.226 | 0.264 | 0.224 | 0.262 | 0.232 | 0.294 | 0.242 | 0.293 |
| Electricity | **0.145** | **0.237** | 0.152 | 0.247 | 0.157 | 0.248 | 0.157 | 0.254 | 0.155 | 0.246 | 0.163 | 0.258 | 0.185 | 0.284 | 0.171 | 0.271 | 0.171 | 0.263 | 0.167 | 0.264 |
| Traffic | 0.365 | **0.245** | **0.357** | 0.248 | 0.393 | 0.256 | 0.387 | 0.271 | 0.374 | 0.255 | 0.395 | 0.279 | 0.409 | 0.279 | 0.397 | 0.275 | 0.522 | 0.282 | 0.418 | 0.287 |
| Solar | **0.176** | **0.214** | 0.183 | 0.219 | 0.195 | 0.214 | 0.184 | 0.242 | 0.198 | 0.249 | 0.202 | 0.262 | 0.193 | 0.252 | 0.200 | 0.284 | 0.205 | 0.233 | 0.224 | 0.286 |
| ETTm1 | **0.331** | **0.364** | 0.349 | 0.380 | 0.338 | 0.369 | 0.348 | 0.380 | 0.359 | 0.382 | 0.361 | 0.390 | 0.356 | 0.380 | 0.349 | 0.381 | 0.464 | 0.456 | 0.356 | 0.379 |
| ETTm2 | **0.243** | **0.301** | 0.247 | 0.305 | 0.248 | 0.308 | 0.250 | 0.312 | 0.271 | 0.322 | 0.269 | 0.327 | 0.257 | 0.318 | 0.256 | 0.314 | 0.501 | 0.505 | 0.259 | 0.325 |
| ETTh1 | **0.391** | **0.416** | 0.401 | 0.426 | 0.401 | 0.420 | 0.430 | 0.441 | 0.409 | 0.430 | 0.439 | 0.448 | 0.427 | 0.441 | 0.419 | 0.436 | 0.439 | 0.461 | 0.425 | 0.439 |
| ETTh2 | **0.327** | **0.379** | 0.345 | 0.386 | 0.336 | 0.385 | 0.345 | 0.395 | 0.352 | 0.402 | 0.370 | 0.403 | 0.349 | 0.397 | 0.351 | 0.395 | 0.894 | 0.680 | 0.470 | 0.468 |

## 3.1. Main Result

The results, presented in Table 1, demonstrate a critical finding: a well-executed, straightforward MLP-based architecture can still outperform complex state-of-the-art models. Notably, we achieve these gains without relying on intricate mechanisms such as multi-resolution and periodicity. By focusing instead on fundamentally improving the forecasting principles, our MLP model-ReNF sets a new state-of-the-art. Overall, ReNF surpasses all leading 2024 models by a significant margin and outperforms even the very competitive 2025 SOTA methods in almost all cases. This manifestation of contrast provides strong evidence for the effectiveness of our proposed techniques.

*Table 2.* Efficiency comparison of ReNF, TimeBridge, and DUET under their optimal configs on each dataset. All metrics are averaged across the four prediction lengths.

| | Model | ReNF | TimeBridge | DUET |
|---|---|---|---|---|
| Weather | Params (MB) | **0.200** | 0.887 | 4.128 |
| | FLOPs (GB) | **0.004** | 4.262 | 0.143 |
| ETTm2 | Params (MB) | **0.393** | 0.460 | 4.658 |
| | FLOPs (GB) | **0.003** | 1.604 | 0.051 |
| Traffic | Param (MB) | 21.476 | 12.431 | **9.910** |
| | FLOPs (GB) | 22.813 | 479.231 | **15.575** |

However, we do not claim that existing specialized techniques are redundant. Rather, as discussed in Sec.2.5, many of these architectural designs were evaluated within unstable training frameworks, suggesting that their true effects may need tedious re-evaluation, which may be a promising future work. For instance, on the Traffic dataset, while ReNF shows significant improvement over many baselines, it does not surpass the MSE score of the large Transformer-based model, TimeBridge. We compare the model complexity between the most competitive models in the Table 2, showing

that ReNF reduces nearly 20x the computational complexity in FLOPs compared to TimeBridge with the Traffic dataset. This suggests that more complex models might still work better on datasets with high volume.

## 3.2. Ablation Study

**Effect of EMA.** To analyze the impact of EMA, we recorded the evaluation dynamics of ReNF with and without our smoothing technique. The results, shown in Fig. 5, clearly demonstrate EMA's role. It is notable that we also record the variation of test loss as evidence for the effect of improving the generalization ability.

First, on smaller or less stable datasets like ETTh2, EMA mitigates spurious overfitting and stabilizes the learning curves. This provides a more reliable signal for early stopping and prolongs the effective training period. Second, on large, high-quality datasets such as Electricity, the choice of EMA decay rate can influence performance. By selecting an appropriate rate, the generalization ability of the NF can be substantially enhanced. Full numerical results for all datasets are available in the Appendix E.4. These results show clearly the advantages of leveraging the model weight averaging technique and strongly support our claim in addressing the issue of valid-test inconsistency when training a neural forecaster for time series.

**Effect of BDO.** We investigate the effect of the BDO paradigm with the following settings: 1) Keep the depth of ReNF, but disable the recursive input concatenation and apply the loss function only to the final output; 2) Directly change the number of layers/sub-forecasts. By config.1, the BDO reduces to DO with the model structure unchanged. The results, shown in Fig. 6, demonstrate that even on the ETTh1 dataset, which has limited data and is difficult to optimize with deep representations, our BDO strategy can still yield superior performance, especially for very long-term forecasts. This finding empirically supports our

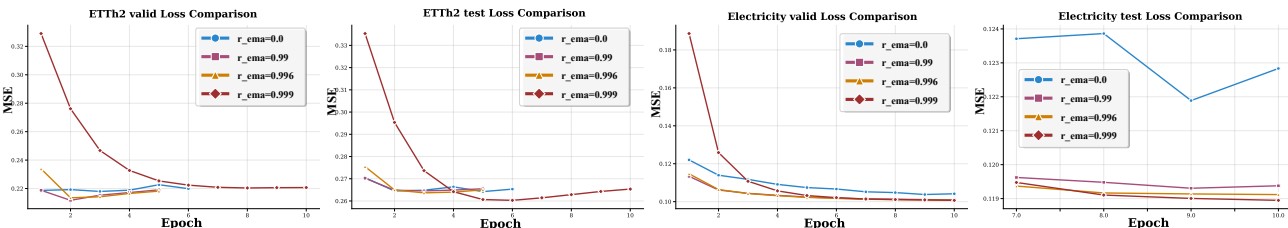

*Figure 5.* Effect of the EMA smoothing on training and test phase. "r_ema" denotes the applied ratio $\alpha$ of EMA weighting.

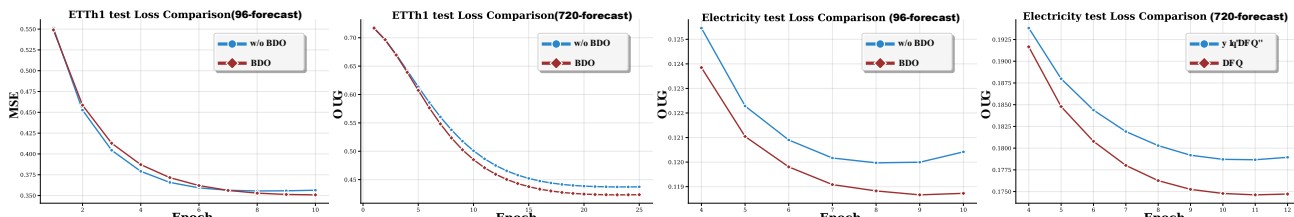

*Figure 6.* Effect of the BDO forecast.

claim, derived from hypothesis 2.1, that stacking multiple, hierarchically-generated forecasts provides useful mutual information that enhances the final combined prediction.

Furthermore, it is shown clearly in Fig.7(a) that the performance of ReNF with the BDO strategy can consistently improve as $K$ increases. In stark contrast, the performance of the DO model stagnates or degrades with added depth. This difference highlights BDO's ability to effectively utilize deeper architectures. This meaningful phenomenon suggests that our paradigm may unlock a new, more effective scaling potential for NFs. Diverse examples are shown in the Table 3 and the Appendix D.

*Table 3.* 720-length forecast with varying number of sub-forecasts.

| Layer | | K=1 | K=2 | K=3 | K=4 | K=5 | K=6 |
|---|---|---|---|---|---|---|---|
| Weather | MSE | 0.311 | 0.309 | 0.307 | 0.307 | 0.307 | 0.307 |
| | MAE | 0.322 | 0.320 | 0.319 | 0.319 | 0.319 | 0.319 |
| ETTm1 | MSE | 0.411 | 0.408 | 0.407 | 0.406 | 0.404 | 0.401 |
| | MAE | 0.411 | 0.409 | 0.408 | 0.407 | 0.407 | 0.407 |
| Traffic | MSE | 0.426 | 0.415 | 0.410 | 0.406 | 0.403 | 0.402 |
| | MAE | 0.281 | 0.274 | 0.272 | 0.270 | 0.267 | 0.267 |

### 3.3. Empirical Evaluation of the Hypothesis

*Table 4.* Comparison between the last single forecast and the optimal post-combined forecast of ReNF. The shown metrics are MSE and are averaged across the four prediction lengths.

| Dataset | | ETTm1 | ETTm2 | ETTh1 | ETTh2 | Weat. | ELC | Solar | Traffic |
|---|---|---|---|---|---|---|---|---|---|
| ReNF | Last Forecast | 0.331 | 0.243 | 0.391 | 0.327 | 0.214 | 0.145 | 0.176 | 0.365 |
| | Empirical Bound | 0.225 | 0.201 | 0.270 | 0.252 | 0.165 | 0.100 | 0.090 | 0.254 |

To empirically evaluate our variance reduction hypothesis, we implement the post-combination strategy described in Sec. 2.1. This requires a combination functional, $g_c$. While finding an optimal solution in practice is difficult, we can establish a theoretical upper bound on performance for the use of evaluation. Given the ground truth labels, a trivial yet optimal strategy is to select, at each timestep, the forecast value from among all candidate forecasts that has the smallest error. By applying this "oracle" combination, we aim to explore two points: 1) to quantify the potential accuracy improvement achievable through post-combination, thereby establishing a dynamic empirical bound for our forecasting model; and 2) to verify that this empirical bound behaves in a manner consistent with the intuitions of the VRH (2.1).

Table 4 presents the performance of this oracle post-combination strategy, revealing a significant gap between the final ReNF forecast and the empirically optimal combination. While this oracle performance is unattainable in practice, the resulting empirical bound is informative. On one hand, it indicates that any single forecast is suboptimal and that powerful combinatorial strategies for improving predictions will probably exist (a promising direction for future work). On the other hand, it shows that we are still far from perfectly leveraging the information contained within the multiple sub-forecasts. This gap precisely highlights the ensemble principle of our BDO paradigm, which is designed to help the neural network implicitly learn a more effective combination functional $g_c$.

Furthermore, Fig. 7(b) shows how this empirical bound varies with the number of sub-forecasts $K$. The bound consistently decays as $K$ increases, even when the model's performance saturates. This result aligns well with the core insight from VRH: the inherent predictive variance reduces,

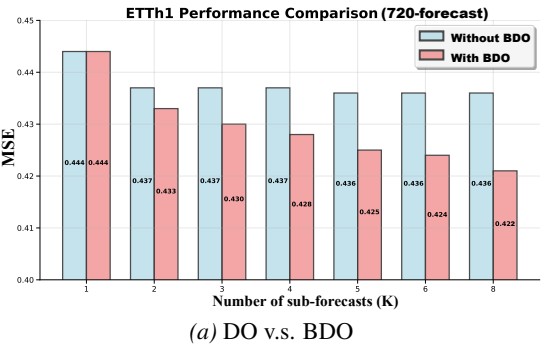

*(a)* DO v.s. BDO

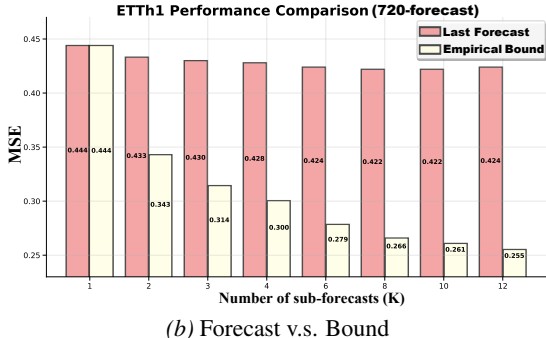

*(b)* Forecast v.s. Bound

*Figure 7.* Performance variation with different numbers of sub-forecasts K. (a). Comparison of using DO or BDO. (b).Comparison between the last forecast and the empirical bound of ReNF.

and the theoretical performance limit improves as the number of candidate forecasts grows.

In a nutshell, our evaluation of the VRH indicates two primary roles for building NNs in this field. First, NNs can be leveraged to learn a powerful meta combination functional over multiple candidate forecasts. Second, stronger base forecasters should be developed to better approximate the true data-generating distribution, directly reducing the predictive bias $b$ as identified in our hypothesis.

## 4. Related Work

**Ensemble.** Our work connects to classic ensemble methods in machine learning (Mohri et al., 2018). For instance, the recursive workflow of ReNF is analogous to gradient-boosted regression models (Chen & Guestrin, 2016), which construct a strong predictor from multiple weak ones. Additionally, the collaborative training can be viewed as a form of bagging (Breiman, 1996), which effectively resamples the data to train diverse forecasters. In this context, an even more recent study (Modi & Pan, 2025) re-certified the benefits of such ensemble methods for enhancing Transformer-based NFs, providing further empirical support for the direction of our research.

Forecast combination is a classic ensemble technique for improving forecast accuracy and robustness by leveraging the diverse strengths of multiple models (Clemen, 1989). In this work, we revisit this concept (Bates & Granger, 1969) in the domain of deep learning for long-term time series forecasting. Rather than combining distinct, parallel forecasters into a hybrid model (Zhang, 2003; Montero-Manso et al., 2020), our framework achieves this goal efficiently within a single, structured approach for generating and implicitly combining forecasts within any neural network.

Our model structure shares a conceptual lineage with deep stacking architectures (Wolpert, 1992) like N-BEATS (Oreshkin et al., 2019) and N-HiTS (Challu et al., 2023), which

also synthesize a final forecast from multiple sub-modules. However, the fundamental motivation and aggregation mechanism differ. While N-BEATS and N-HiTS rely on additive residual decomposition—where subsequent blocks focus on correcting the errors (or specific frequency components) of previous blocks—our BDO paradigm employs recursive input concatenation. Instead of summing independent components, ReNF generates forecasts in a curriculum-style progression (from short-term to long-term horizons). By explicitly feeding earlier sub-forecasts back into the input space of subsequent layers, ReNF enforces a sequential dependency and allows the model to iteratively refine its understanding of the temporal trajectory, rather than performing a static multi-scale decomposition.

**Iterative Refinement.** Our recursive approach, BDO, also shares conceptual similarities with iterative refinement and self-conditioning techniques used in Natural Language Processing and Reinforcement Learning (Madaan et al., 2023; Du et al., 2022). However, while standard iterative refinement focuses on progressively improving a fixed-length output, BDO does not constrain subsequent steps to be "better" versions of the previous output; instead, each step expands the forecasting horizon while implicitly combining prior sub-forecasts, explicitly injecting causal structure into long-term forecasting.

**Short to Long Curriculum.** We note that ReNF is not the first work to address LTSF using multiple short-term forecasts. A pioneering and relevant study by (Nguyen & Chan, 2004) proposed a machine called MNN, which generates long-term forecasts using multiple neural networks, each responsible for a specific interval of the output. The essential distinction between MNN and our approach is that we generate sub-forecasts recursively, with the explicit goal of injecting structural causality into the DO strategy. In contrast, each sub-network in MNN functions as a standalone AR(1) model. This difference stems from our distinct motivation: whereas MNN was designed primarily to mitigate the error accumulation of one-step-ahead NN models, our

work begins with a foundational hypothesis. Based on this hypothesis, we leverage modern deep learning techniques to effectively utilize multiple sub-forecasts, achieving strong empirical results on a wide range of real-world datasets, which constitutes one of our main contributions.

## 5. Limitations and Further Discussion

The Variance Reduction Hypothesis (VRH) presented in this paper is preliminary, as it relies on a strict independence (i.i.d.) assumption. As discussed, its primary role is to serve as a motivating heuristic and core intuition for our multiple-forecasting design, rather than a strict theoretical proof. Nevertheless, a more rigorous exploration of its theoretical properties could inspire promising research directions. Specifically, as shown in Section 3.3, increasing the number of sub-forecasts consistently reduces the combined forecasting error, indicating that the core intuition of VRH remains conceptually valid even when strict independence is relaxed. Therefore, future work could explore the error covariance between sub-forecasts to derive a theoretical bound that accounts for non-independent forecasts. This would better align with our proposal and identify more intricate improvements to the current structure.

Furthermore, our employment of the Exponential Moving Average (EMA) to mitigate training instability caused by non-stationary time series represents an initial exploration. While EMA successfully smooths the validation signal, prevents premature early stopping, and allows for reliable model evaluation, the fundamental reasons why it yields such significant accuracy improvements specifically for MLP architectures are not yet comprehensively understood. Future research should investigate whether more intricate weight-averaging methods could yield broader benefits and explore the precise mechanisms through which optimization smoothing interacts with diverse neural architectures in time series forecasting.

Under the paradigm of VRH, the role of Neural Network (NN) in this area becomes transparent. First and perhaps most importantly, we should leverage NN's capability to create a powerful post-combination function. While our BDO paradigm is effective, the current recursive strategy for combining sub-forecasts is not fully optimal. This is evident from the performance gap between our final forecast and the theoretical empirical bound as the number of stages increases. The core is that while BDO has created the skeleton of sequential causality at the level of network structure, methods for explicitly enforcing the specific causality in this paradigm are required. Developing intricate methods to leverage the full set of sub-forecasts better could lead to substantial accuracy gains. Second, we are consistently supposed to build more powerful Neural Forecasting Machines to approximate the expected distributions of future data, thereby reducing the bias $b$ in the VRH B.1. Therefore, a comprehensive study is needed to verify the effects of our proposed techniques when applied to other advanced model architectures beyond MLP.

Finally, although we show the potential on diverse short-term forecast benchmarks, this work develops the BDO paradigm specifically for the LTSF setting. A key open question is how this paradigm can be better adapted for diverse forecasting tasks with short length, which could ultimately lead to a more unified framework for time series forecasting.

## 6. Conclusion

We diagnose some fundamental issues in the current design of neural forecasters. Driven by the Variance Reduction Hypothesis conceptually, and the lack of structural causality in Direct Output models practically, we introduce the Boosted Direct Output framework. This paradigm implicitly realizes forecast combination within a causal, hierarchical representation space. We identify parameter smoothing (e.g., EMA) as a critical, yet often overlooked, mechanism for resolving the validation-test generalization gap in non-stationary time series. Extensive experiments demonstrate that these principled improvements enable a simple MLP to surpass recent complex state-of-the-art models. Finally, the value of our hypothesis is evaluated and confirmed by experimental results, identifying the role of the neural network in building the Long-Term Time Series forecasters.

## Acknowledgements

This work was funded by the Anhui Provincial Natural Science Foundation under Grant 2308085US01, the Key Science & Technology Project of Anhui Province under Grant 202523j08050017, the Major Project for Beidou Industrialization of the National Development and Reform Commission under Grant NDRC Investment [2023] No. 849, and the Director's Fund of Hefei Institutes of Physical Science, Chinese Academy of Sciences under Grant YZJJ202301-CX. We are grateful to the reviewers for their critical comments.

## Impact Statement

This research highlights the critical importance of foundational design over architectural complexity in long-term time series forecasting. We demonstrate that bridging stabilization techniques (e.g., weight averaging) with an improved forecasting structure, in the spirit of classic forecast combination, enables a simple MLP to outperform heavily parameterized state-of-the-art models. This may encourage the community to reconsider fundamental forecasting principles rather than relying solely on architectural engineering.

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

# Appendix

## List of Appendices:

## A. Details of Datasets

The ETTh1, ETTh2, ETTm1, and ETTm2 datasets record the temperature of electricity transformers every hour and every 15 minutes. The Weather dataset contains 21 pieces of weather information, measured every 10 minutes in Germany. The Electricity dataset records the amount of electricity used by 321 customers every hour. The Solar dataset records how much electricity is produced by solar power stations every 10 minutes, from 137 solar power stations, in 2006. The Traffic dataset records how busy the roads are in San Francisco, every hour, from 862 sensors on the freeway.

In addition, we also evaluate the short-term time series forecasting on eight multivariate datasets that are used in the study (Yue et al., 2025). The METR-LA database was populated with traffic network data for Los Angeles during the spring of 2012, specifically from March to June. This data was collected at an interval of five minutes. The NASDAQ includes the daily NASDAQ index and key economic indicators from 2010 to 2024. The SP500 records daily SP500 index data (e.g., opening price, closing price, and trading volume) from January 1993 to February 2025. CarSales collects daily sales of 10 vehicle brands (e.g., Toyota, Honda) in the U.S. from January 2005 to June 2023. The data are compiled from the Vehicle Sales dataset on Kaggle. Power contains daily wind and solar energy production (in MW) records for the French grid from April 2020 to June 2023. The data are compiled from the Wind and Solar Daily Power Production dataset on Kaggle. The website contains six years of daily visit data (e.g, first-time and returning visits) to an academic website, spanning from September 2014 to August 2020.

*Table 5.* Descriptions of multivariate time series datasets used in the table 6 and table 7. The Variate column represents the number of variates, and the split column specifies the train-validate-test splitting ratio for each dataset.

| Dataset | Variate | Prediction Length | Split | Frequency | domain |
|---|---|---|---|---|---|
| ETTh1, ETTh2 | 7 | 96, 192, 336, 720 | (6, 2, 2) | Hourly | Electricity |
| ETTm1, ETTm2 | 7 | 96, 192, 336, 720 | (6, 2, 2) | 15min | Electricity |
| Weather | 21 | 96, 192, 336, 720 | (7, 1, 2) | 10min | Environment |
| Electricity | 321 | 96, 192, 336, 720 | (7, 1, 2) | Hourly | Electricity |
| Traffic | 862 | 96, 192, 336, 720 | (7, 1, 2) | Hourly | Transportation |
| Solar | 137 | 96, 192, 336, 720 | (6, 2, 2) | 10min | Energy |
| NASDAQ | 12 | 24, 36, 48, 60 | (7, 1, 2) | Daily | Finance |
| SP500 | 12 | 24, 36, 48, 60 | (7, 1, 2) | Daily | Finance |
| Carsales | 12 | 24, 36, 48, 60 | (7, 1, 2) | Daily | Market |
| Website | 12 | 24, 36, 48, 60 | (7, 1, 2) | Daily | Web |
| Power | 12 | 24, 36, 48, 60 | (7, 1, 2) | Daily | Energy |
| METR-LA | 207 | 24, 36, 48, 60 | (7, 1, 2) | 5min | Transportation |

# B. Theoretical Analysis

## B.1. Proof of the Hypothesis

*Proof.* First, we derive the expected bound for each element $y_t^{(i)}$ in a candidate forecast $\hat{Y}_f^{(i)} = \left\{ y_1^{(i)}, y_2^{(i)}, \cdots, y_c^{(i)} \right\}$, by assuming that each candidate forecast at a fixed time $t$, i.e., $\{y_t^{(i)}\}_{i=1}^c$, are independent samples from the same distribution with $\mathbb{E}[y_t] = \hat{\mu}_t$.

In particular, we have

$$\mathbb{E}\left| \hat{\mu}_t - \frac{1}{c} \sum_{i=1}^c y_t^{(i)} \right|^2 = \mathbb{E}\left| \frac{1}{c} \sum_{i=1}^c (y_t^{(i)} - \hat{\mu}_t) + \hat{\mu}_t - \hat{\mu}_t \right|^2 \tag{6}$$

$$= \frac{1}{c^2} \mathbb{E}\left| \sum_{i=1}^c (y_t^{(i)} - \hat{\mu}_t) \right|^2 \tag{7}$$

$$= \frac{1}{c^2} \sum_{i=1}^c \mathbb{E}\left| y_t^{(i)} - \hat{\mu}_t \right|^2. \tag{8}$$

The last identity holds because $\{y_t^{(i)}\}_{i=1}^c$ are independent, i.e., $\mathbb{E}[(y_t^{(i)} - \hat{\mu}_t)(y_t^{(j)} - \hat{\mu}_t)] = \mathbb{E}(y_t^{(i)} - \hat{\mu}_t) \cdot \mathbb{E}(y_t^{(j)} - \hat{\mu}_t) = 0, i \neq j$.

Then, we can bound this term by showing

$$\mathbb{E}\left| y_t^{(i)} - \hat{\mu}_t \right|^2 = \mathbb{E}\left| y_t^{(i)} - \mathbb{E}[y_t^{(i)}] \right|^2 \tag{9}$$

$$= \mathbb{E}\left| (y_t^{(i)})^2 \right| - \left| \mathbb{E}[y_t^{(i)}] \right|^2 \quad \text{(variance identity)} \tag{10}$$

$$\leq \mathbb{E}[(y_t^{(i)})^2] \leq \lambda^2. \tag{11}$$

where $\lambda$ represents the upper bound of the second moment $\mathbb{E}[(y_t^{(i)})^2]$.

Therefore, we have shown that

$$\mathbb{E}\left|\hat{\mu}_t - \frac{1}{c}\sum_{i=1}^{c} y_t^{(i)}\right|^2 \leq \frac{\lambda^2}{c}. \tag{12}$$

for any $i$; By Jensen's inequality, we bound the expected absolute deviation

$$\mathbb{E}\left|\hat{\mu}_t - \frac{1}{c}\sum_{i=1}^{c} y_t^{(i)}\right| \leq \frac{\lambda}{\sqrt{c}}. \tag{13}$$

Now, we bound the total expected error between the observation $x_t$ and the ensemble average. By the triangle inequality

$$\left|x_t - \frac{1}{c}\sum_{i=1}^{c} y_t^{(i)}\right| \leq \underbrace{|x_t - \mu_t|}_{\text{Noise}} + \underbrace{|\mu_t - \hat{\mu}_t|}_{\text{Bias}} + \underbrace{\left|\hat{\mu}_t - \frac{1}{c}\sum_{i=1}^{c} y_t^{(i)}\right|}_{\text{Variance}}. \tag{14}$$

Taking the expectation of both sides, and assuming the expected model bias is bounded by $\mathbb{E}|\mu_t - \hat{\mu}_t| \leq b$ and the expected intrinsic noise is bounded by $\mathbb{E}|x_t - \mu_t| \leq \sigma$ (which is reasonable with the goal to minimize the Mean Absolute Error (MAE) or Mean Squared Error (MSE)), we have

$$\mathbb{E}\left|x_t - \frac{1}{c}\sum_{i=1}^{c} y_t^{(i)}\right| \leq \sigma + b + \frac{\lambda}{\sqrt{c}}. \tag{15}$$

Summing over the horizon $t = 1$ to $L$, we derive the final expected upper bound

$$\sum_{t=1}^{L} \mathbb{E}\left|x_t - \frac{1}{c}\sum_{i=1}^{c} y_t^{(i)}\right| \leq L\left(\frac{\lambda}{\sqrt{c}} + b + \sigma\right). \tag{16}$$

This completes the proof.

## B.2. Variance Analysis of Total Summed Error

The above proof gives an intuitive bound, which does not depend on any assumption on the temporal dependence. In this part, we consider this factor as a pivot for analysing the effects of different forecasting paradigms, identifying the role of our BDO.

The variance of this total error is:

$$var\left(\sum_{t=1}^{T} e_t\right) = \sum_{t=1}^{T} var(e_t) + 2\sum_{t<t'} Cov(e_t, e'_t) \tag{17}$$

For simplicity, let the prediction error at each timestep be $t$ $e_t := \mu_t - \frac{1}{c}\sum_{i=1}^{c} \tilde{y}_t^{(i)}$, and temporarily omit the effect of the predictive bias $b$ of NFM, i.e., $\mu = \hat{\mu}$. Then we can compute the covariance (autocorrelation) of the errors at any two distinct timesteps $k$ and $h$,

$$Cov(e_k, e_h) = \mathbb{E}\left[(\mu_k - \tilde{\mu}_k)(\mu_h - \tilde{\mu}_h)\right] \tag{18}$$
$$= \mathbb{E}[\tilde{\mu}_k\tilde{\mu}_h] - \mu_k\mu_h \tag{19}$$
$$= Cov(\tilde{\mu}_k\tilde{\mu}_h) \tag{20}$$

where we denote $\tilde{\mu}_t := \frac{1}{c}\sum_{i=1}^{c} \tilde{y}_t^{(i)}$. So we have

$$Cov(\tilde{\mu}_k\tilde{\mu}_h) = \frac{1}{c^2}\sum_{i=1}^{c}\sum_{j=1}^{c} Cov(y_k^{(i)}, y_h^{(j)}) \tag{21}$$

$$= \frac{1}{c} Cov(y_k^{(i)}, y_h^{(i)}), \ \forall i \tag{22}$$

The last equality holds by isolating the correlations across different candidates.

Now we can consider the role of temporal dependence in the error structure:

**Case 1: Temporal Independence (The DO Paradigm):** The DO paradigm is designed to (approximately) satisfy the assumption of temporal independence, as evidenced in Sec. 2. This implies that the error covariance terms in Eq. 22 are (nearly) zero. Consequently, the total error variance is simply the sum of the per-timestep variances, While this structure effectively prevents the compounding of errors, it often comes at the cost of the model being "unaware" of the sequential dynamics within the future horizon, potentially leading to higher per-timestep variance.

**Case 2: Temporal Dependence (The AR Paradigm):** The typical AR paradigm fundamentally violates this assumption and thus leads to errors from one step propagating to the next, which causes the total variance to explode over long horizons. Even though the introduction of the ensembling factor c would reduce the magnitude of this effect, it does not alter the underlying structural problem of error accumulation.

**Case 3: A Synthesis (The BDO Paradigm):** Our Boosted Direct Output paradigm operates as a synthesis of these two extremes. It intentionally violates strict temporal independence by recursively feeding sub-forecasts back into the model, thereby making it aware of causal dependencies within the forecast horizon. However, unlike a pure AR model, through hierarchical supervision and its patch-wise output structure inherited from DO, the model is trained to control the accumulation of errors. The goal is to leverage the benefits of modeling temporal structure while constraining the error correlation, effectively learning to make the covariance term in Eq. 22 as small as possible.

Note that the above proof and analysis are based on a univariate time series; however, it is easy to extend the result to the multivariate case using a similar process in a certain normed vector space.

## C. Implicit Structural Loss

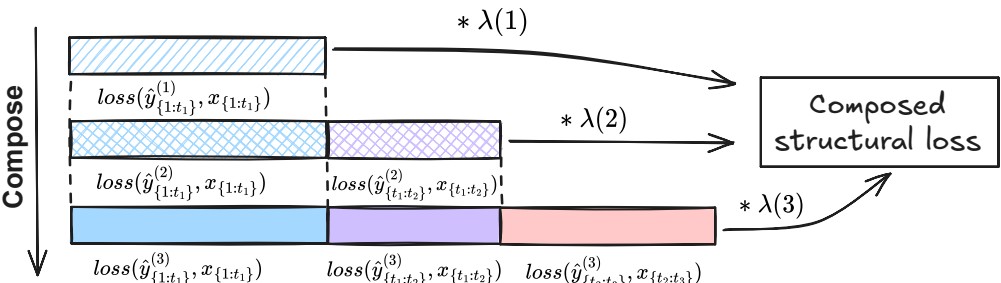

*Figure 8.* Illustration of the implicit structural loss of BDO, we exemplify it in the NF consisting of three sub-forecasters.

An interesting property of the BDO paradigm is the implicit structural loss it induces. This connects our work to recent research on explicit loss engineering, such as positional weighting (Xue et al., 2024) and patch-wise structural losses (Kudrat et al., 2025). We posit that the BDO learning objective, formed by the weighted sum of losses from hierarchical sub-forecasts, inherently functions as a complex structural loss. This seemingly implicit loss can be seen as the generalized version of the above two.

To formalize this property, we first simplify the loss function from Eq.4 as:

$$Loss = \sum_{n=1}^{N} \lambda(n) * f(\hat{Y}_f^{(n)}, X_f^{(n)}) \tag{23}$$

where $f(\cdot, \cdot)$ denotes a base error function such as the MAE.

Note that in our definitions and notations, $\hat{Y}_f^{(n)} = \{\hat{y}_1^{(n)}, \hat{y}_2^{(n)}, \cdots, \hat{y}_{t_n}^{(n)}\}$. We can therefore expand the total loss into a point-wise sum over all timesteps:

$$Loss = \sum_{n=1}^{N} \sum_{t=1}^{t_n} \lambda(n) * f(\hat{y}_t^{(n)}, x_t) \tag{24}$$

The underlying structure of this composite loss, visualized in Figure 8, reveals a key insight: BDO is not just a recursive forecasting process, but also a method for implicitly constructing a complex and adaptive structural loss. The properties of this loss can be finely tuned through several mechanisms: the stage-wise weighting coefficients, the forecast splitting strategy, the choice of the base error function, and even the architecture of each sub-forecaster.

## D. Full Empirical Results

### D.1. Long-term Forecasting Results

We present the full version of Table.1 in Table.6 to show the performance of ReNF in Long-Term time series forecasting.

*Table 6.* Full results of long-term forecasting of hyperparameter searching. The Emp-Bound column denotes the empirical bound discussed in Sec.3.3. The look-back window is searched from {336, 512, 720} for the best performance. Timer-XL uses a 672-length window for the best performance, as shown in the original paper. All results are averaged across four different prediction lengths: {96, 192, 336, 720}. The **best** and second-best results are highlighted.

| Models | Emp Bound - | | ReNF ours | | TimeBridge ((2025a)) | | DUET (2025) | | TimeDistill (2026) | | Timer-XL (2025b) | | iTransformer (2024) | | TimeMixer (2024b) | | PatchTST (2023) | | Crossformer (2023) | | DLinear (2023) | |
|---|---|---|---|---|---|---|---|---|---|---|---|---|---|---|---|---|---|---|---|---|---|---|---|
| Metric | MSE | MAE | MSE | MAE | MSE | MAE | MSE | MAE | MSE | MAE | MSE | MAE | MSE | MAE | MSE | MAE | MSE | MAE | MSE | MAE | MSE | MAE |
| **Weather** 96 | 0.110 | 0.149 | **0.138** | **0.180** | 0.144 | 0.184 | 0.146 | 0.191 | 0.145 | 0.204 | 0.157 | 0.205 | 0.157 | 0.206 | 0.147 | 0.198 | 0.150 | 0.200 | 0.143 | 0.210 | 0.170 | 0.230 |
| 192 | 0.141 | 0.185 | **0.182** | **0.224** | 0.186 | 0.226 | 0.188 | 0.231 | 0.188 | 0.247 | 0.206 | 0.250 | 0.200 | 0.248 | 0.192 | 0.242 | 0.191 | 0.239 | 0.195 | 0.261 | 0.216 | 0.273 |
| 336 | 0.177 | 0.216 | **0.231** | **0.266** | 0.237 | 0.267 | 0.235 | 0.269 | 0.240 | 0.286 | 0.259 | 0.291 | 0.252 | 0.287 | 0.247 | 0.284 | 0.242 | 0.279 | 0.254 | 0.319 | 258 | 0.307 |
| 720 | 0.232 | 0.261 | **0.304** | **0.318** | 0.312 | 0.321 | 0.308 | 0.319 | 0.310 | 0.338 | 0.337 | 0.344 | 0.320 | 0.336 | 0.318 | 0.330 | 0.312 | 0.330 | 0.335 | 0.385 | 0.323 | 0.362 |
| *Avg.* | 0.165 | 0.203 | **0.214** | **0.247** | 0.220 | 0.250 | 0.219 | 0.253 | 0.221 | 0.269 | 0.240 | 0.273 | 0.232 | 0.269 | 0.226 | 0.264 | 0.224 | 0.262 | 0.232 | 0.294 | 0.242 | 0.293 |
| **Electricity** 96 | 0.083 | 0.171 | **0.118** | **0.210** | 0.122 | 0.216 | 0.128 | 0.219 | 0.128 | 0.225 | 0.127 | 0.219 | 0.134 | 0.230 | 0.153 | 0.256 | 0.143 | 0.247 | 0.134 | 0.231 | 0.140 | 0.237 |
| 192 | 0.097 | 0.186 | **0.138** | **0.229** | 0.143 | 0.238 | 0.145 | 0.235 | 0.145 | 0.241 | 0.145 | 0.236 | 0.154 | 0.250 | 0.168 | 0.269 | 0.158 | 0.261 | 0.146 | 0.243 | 0.154 | 0.251 |
| 336 | 0.105 | 0.196 | **0.151** | **0.244** | 0.163 | 0.258 | 0.163 | 0.255 | 0.161 | 0.258 | 0.159 | 0.252 | 0.169 | 0.265 | 0.189 | 0.291 | 0.168 | 0.267 | 0.165 | 0.264 | 0.169 | 0.268 |
| 720 | 0.115 | 0.207 | **0.173** | **0.266** | 0.178 | 0.274 | 0.193 | 0.281 | 0.195 | 0.291 | 0.187 | 0.277 | 0.194 | 0.288 | 0.228 | 0.320 | 0.214 | 0.307 | 0.237 | 0.314 | 0.204 | 0.301 |
| *Avg.* | 0.100 | 0.190 | **0.145** | **0.237** | 0.152 | 0.247 | 0.157 | 0.248 | 0.157 | 0.254 | 0.155 | 0.246 | 0.163 | 0.258 | 0.185 | 0.284 | 0.171 | 0.271 | 0.171 | 0.263 | 0.167 | 0.264 |
| **Traffic** 96 | 0.234 | 0.167 | 0.335 | **0.226** | 0.332 | 0.237 | 0.360 | 0.238 | 0.358 | 0.256 | 0.340 | 0.238 | 0.358 | 0.258 | 0.369 | 0.257 | 0.370 | 0.262 | 0.526 | 0.288 | 0.395 | 0.275 |
| 192 | 0.245 | 0.175 | 0.356 | 0.239 | **0.343** | **0.239** | 0.383 | 0.249 | 0.374 | 0.264 | 0.360 | 0.248 | 0.382 | 0.271 | 0.399 | 0.272 | 0.386 | 0.269 | 0.503 | 0.263 | 0.407 | 0.280 |
| 336 | 0.254 | 0.180 | 0.366 | **0.246** | **0.360** | 0.249 | 0.395 | 0.259 | 0.389 | 0.271 | 0.377 | 0.256 | 0.396 | 0.277 | 0.407 | 0.272 | 0.396 | 0.275 | 0.505 | 0.276 | 0.417 | 0.286 |
| 720 | 0.284 | 0.197 | 0.402 | **0.267** | **0.392** | 0.268 | 0.435 | 0.278 | 0.428 | 0.292 | 0.418 | 0.279 | 0.445 | 0.308 | 0.461 | 0.316 | 0.435 | 0.295 | 0.552 | 0.301 | 0.454 | 0.308 |
| *Avg.* | 0.254 | 0.180 | 0.365 | **0.245** | **0.357** | 0.248 | 0.393 | 0.256 | 0.387 | 0.271 | 0.374 | 0.255 | 0.395 | 0.279 | 0.409 | 0.279 | 0.397 | 0.275 | 0.522 | 0.282 | 0.418 | 0.287 |
| **Solar** 96 | 0.082 | 0.141 | **0.157** | 0.202 | 0.159 | **0.196** | 0.166 | 0.211 | 0.166 | 0.229 | 0.162 | 0.221 | 0.190 | 0.244 | 0.179 | 0.232 | 0.170 | 0.234 | 0.183 | 0.208 | 0.199 | 0.265 |
| 192 | 0.087 | 0.145 | 0.174 | **0.210** | **0.173** | 0.215 | 0.199 | 0.212 | 0.181 | 0.239 | 0.187 | 0.239 | 0.193 | 0.257 | 0.201 | 0.259 | 0.204 | 0.302 | 0.208 | 0.227 | 0.220 | 0.281 |
| 336 | 0.099 | 0.156 | **0.180** | 0.219 | 0.193 | 0.229 | 0.207 | **0.215** | 0.191 | 0.246 | 0.205 | 0.255 | 0.203 | 0.266 | 0.190 | 0.256 | 0.212 | 0.293 | 0.212 | 0.239 | 0.234 | 0.295 |
| 720 | 0.092 | 0.151 | **0.190** | 0.225 | 0.207 | 0.237 | 0.206 | **0.217** | 0.199 | 0.252 | 0.238 | 0.279 | 0.223 | 0.281 | 0.203 | 0.261 | 0.215 | 0.307 | 0.215 | 0.256 | 0.243 | 0.301 |
| *Avg.* | 0.090 | 0.148 | **0.176** | **0.214** | 0.183 | 0.219 | 0.195 | 0.214 | 0.184 | 0.242 | 0.198 | 0.249 | 0.202 | 0.262 | 0.193 | 0.252 | 0.200 | 0.284 | 0.205 | 0.233 | 0.224 | 0.286 |
| **ETTm1** 96 | 0.189 | 0.262 | **0.270** | **0.325** | 0.288 | 0.339 | 0.279 | 0.333 | 0.285 | 0.344 | 0.290 | 0.341 | 0.300 | 0.353 | 0.293 | 0.345 | 0.289 | 0.342 | 0.314 | 0.367 | 0.300 | 0.345 |
| 192 | 0.218 | 0.284 | **0.310** | **0.352** | 0.326 | 0.368 | 0.320 | 0.358 | 0.331 | 0.368 | 0.337 | 0.369 | 0.341 | 0.380 | 0.335 | 0.372 | 0.329 | 0.368 | 0.374 | 0.410 | 0.336 | 0.366 |
| 336 | 0.242 | 0.300 | **0.343** | **0.373** | 0.363 | 0.394 | 0.348 | 0.377 | 0.359 | 0.386 | 0.374 | 0.393 | 0.374 | 0.396 | 0.368 | 0.386 | 0.362 | 0.390 | 0.413 | 0.432 | 0.367 | 0.387 |
| 720 | 0.249 | 0.296 | **0.400** | **0.405** | 0.417 | 0.419 | 0.405 | 0.408 | 0.415 | 0.416 | 0.437 | 0.428 | 0.429 | 0.430 | 0.426 | 0.417 | 0.416 | 0.423 | 0.753 | 0.613 | 0.419 | 0.417 |
| *Avg.* | 0.225 | 0.286 | **0.331** | **0.364** | 0.349 | 0.380 | 0.338 | 0.369 | 0.348 | 0.380 | 0.359 | 0.382 | 0.361 | 0.390 | 0.356 | 0.380 | 0.349 | 0.381 | 0.464 | 0.456 | 0.356 | 0.379 |
| **ETTm2** 96 | 0.131 | 0.214 | **0.157** | **0.241** | 0.157 | 0.243 | 0.162 | 0.249 | 0.163 | 0.255 | 0.175 | 0.257 | 0.175 | 0.266 | 0.165 | 0.256 | 0.165 | 0.255 | 0.296 | 0.391 | 0.164 | 0.256 |
| 192 | 0.174 | 0.246 | **0.212** | **0.279** | 0.218 | 0.284 | 0.215 | 0.288 | 0.220 | 0.294 | 0.242 | 0.301 | 0.242 | 0.312 | 0.225 | 0.298 | 0.221 | 0.293 | 0.369 | 0.416 | 0.224 | 0.304 |
| 336 | 0.216 | 0.276 | **0.262** | **0.315** | 0.270 | 0.321 | 0.267 | 0.321 | 0.269 | 0.328 | 0.293 | 0.337 | 0.282 | 0.337 | 0.277 | 0.332 | 0.276 | 0.327 | 0.588 | 0.600 | 0.277 | 0.337 |
| 720 | 0.283 | 0.322 | **0.341** | **0.368** | 0.344 | 0.372 | 0.348 | 0.373 | 0.346 | 0.369 | 0.376 | 0.390 | 0.375 | 0.394 | 0.360 | 0.385 | 0.362 | 0.381 | 0.750 | 0.612 | 0.371 | 0.401 |
| *Avg.* | 0.201 | 0.265 | **0.243** | **0.301** | 0.247 | 0.305 | 0.248 | 0.308 | 0.250 | 0.312 | 0.271 | 0.322 | 0.269 | 0.327 | 0.257 | 0.318 | 0.256 | 0.314 | 0.501 | 0.505 | 0.259 | 0.325 |
| **ETTh1** 96 | 0.243 | 0.300 | **0.350** | **0.383** | 0.355 | 0.391 | 0.353 | 0.386 | 0.373 | 0.401 | 0.364 | 0.397 | 0.386 | 0.405 | 0.372 | 0.401 | 0.377 | 0.397 | 0.411 | 0.435 | 0.379 | 0.403 |
| 192 | 0.276 | 0.327 | **0.385** | **0.408** | 0.389 | 0.414 | 0.398 | 0.409 | 0.411 | 0.426 | 0.405 | 0.424 | 0.424 | 0.440 | 0.413 | 0.429 | 0.409 | 0.425 | 0.409 | 0.438 | 0.408 | 0.419 |
| 336 | 0.294 | 0.342 | **0.405** | **0.425** | 0.415 | 0.435 | 0.415 | 0.428 | 0.439 | 0.444 | 0.427 | 0.439 | 0.449 | 0.460 | 0.438 | 0.450 | 0.431 | 0.444 | 0.433 | 0.457 | 0.440 | 440 |
| 720 | 0.266 | 0.322 | **0.422** | **0.449** | 0.443 | 0.462 | 0.436 | 0.458 | 0.495 | 0.493 | 0.439 | 0.459 | 0.495 | 0.487 | 0.486 | 0.484 | 0.457 | 0.477 | 0.501 | 0.514 | 0.471 | 0.493 |
| *Avg.* | 0.270 | 0.323 | **0.391** | **0.416** | 0.401 | 0.426 | 0.401 | 0.420 | 0.430 | 0.441 | 0.409 | 0.430 | 0.439 | 0.448 | 0.427 | 0.441 | 0.419 | 0.436 | 0.439 | 0.461 | 0.425 | 0.439 |
| **ETTh2** 96 | 0.214 | 0.285 | **0.261** | **0.329** | 0.270 | 0.331 | 0.271 | 0.335 | 0.273 | 0.336 | 0.277 | 0.343 | 0.297 | 0.348 | 0.281 | 0.351 | 0.274 | 0.337 | 0.728 | 0.603 | 0.300 | 0.364 |
| 192 | 0.261 | 0.320 | **0.320** | **0.370** | 0.338 | 0.375 | 0.335 | 0.376 | 0.334 | 0.381 | 0.348 | 0.391 | 0.372 | 0.403 | 349 | 0.387 | 0.348 | 0.384 | 0.723 | 0.607 | 0.387 | 0.423 |
| 336 | 0.270 | 0.327 | **0.346** | **0.394** | 0.370 | 0.402 | 0.354 | 0.398 | 0.363 | 0.415 | 0.375 | 0.418 | 0.388 | 0.418 | 0.366 | 0.413 | 0.377 | 0.416 | 0.740 | 0.628 | 0.490 | 0.487 |
| 720 | 0.261 | 0.325 | **0.381** | **0.423** | 0.402 | 0.434 | 0.384 | 0.426 | 0.408 | 0.446 | 0.409 | 0.458 | 0.424 | 0.444 | 0.401 | 0.436 | 0.406 | 0.441 | 1.386 | 0.882 | 0.704 | 0.597 |
| *Avg.* | 0.252 | 0.314 | **0.327** | **0.379** | 0.345 | 0.386 | 0.336 | 0.384 | 0.345 | 0.395 | 0.352 | 0.402 | 0.370 | 0.403 | 0.349 | 0.397 | 0.351 | 0.395 | 0.894 | 0.680 | 0.470 | 0.468 |

## D.2. Short-term Forecasting Results

To further demonstrate the generality of ReNF, we test it at six supplementary datasets from a recent short-term time series forecasting benchmark (Yue et al., 2025). The descriptions of these datasets can be found in the Table.5. As shown in Table

*Table 7.* Full results for the short-term forecasting. We use the look-back window with length $T = 36$ to predict lengths $\{24, 36, 48, 60\}$. The **best results** and second-best results are highlighted.

| Model | | ReNF (Ours) | | OLinear 2025 | | TimeMix. 2024b | | FilterNet 2024 | | FITS 2024 | | DLinear 2023 | | TimeMix.++ 2025 | | Leddam 2024 | | CARD 2024 | | Fredformer 2024 | | iTrans. 2024 | | PatchTST 2023 | |
|---|---|---|---|---|---|---|---|---|---|---|---|---|---|---|---|---|---|---|---|---|---|---|---|---|---|
| Metric | | MSE | MAE | MSE | MAE | MSE | MAE | MSE | MAE | MSE | MAE | MSE | MAE | MSE | MAE | MSE | MAE | MSE | MAE | MSE | MAE | MSE | MAE | MSE | MAE |
| CarSales | 24 | 0.263 | 0.283 | 0.320 | 0.302 | 0.320 | 0.318 | 0.318 | 0.319 | 0.359 | 0.347 | 0.354 | 0.350 | 0.323 | 0.320 | 0.325 | 0.322 | 0.337 | 0.321 | 0.319 | 0.326 | 0.303 | 0.312 | 0.319 | 0.319 |
| | 36 | 0.280 | 0.298 | 0.334 | 0.315 | 0.332 | 0.331 | 0.331 | 0.330 | 0.373 | 0.360 | 0.368 | 0.365 | 0.351 | 0.348 | 0.337 | 0.333 | 0.348 | 0.333 | 0.333 | 0.335 | 0.318 | 0.323 | 0.332 | 0.330 |
| | 48 | 0.299 | 0.315 | 0.347 | 0.327 | 0.345 | 0.343 | 0.342 | 0.341 | 0.385 | 0.370 | 0.382 | 0.379 | 0.351 | 0.342 | 0.351 | 0.346 | 0.362 | 0.345 | 0.349 | 0.344 | 0.331 | 0.332 | 0.347 | 0.344 |
| | 60 | 0.315 | 0.328 | 0.358 | 0.337 | 0.355 | 0.351 | 0.352 | 0.349 | 0.399 | 0.385 | 0.388 | 0.380 | 0.363 | 0.352 | 0.361 | 0.353 | 0.372 | 0.353 | 0.359 | 0.349 | 0.344 | 0.342 | 0.355 | 0.348 |
| | Avg | 0.289 | 0.306 | 0.340 | 0.320 | 0.338 | 0.336 | 0.336 | 0.335 | 0.379 | 0.365 | 0.373 | 0.368 | 0.347 | 0.340 | 0.343 | 0.338 | 0.355 | 0.338 | 0.340 | 0.338 | 0.324 | 0.327 | 0.338 | 0.335 |
| Power | 24 | 1.268 | 0.855 | 1.343 | 0.870 | 1.341 | 0.881 | 1.410 | 0.916 | 1.491 | 0.944 | 1.390 | 0.916 | 1.340 | 0.877 | 1.397 | 0.909 | 1.406 | 0.886 | 1.410 | 0.913 | 1.462 | 0.924 | 1.468 | 0.935 |
| | 36 | 1.336 | 0.880 | 1.445 | 0.903 | 1.420 | 0.914 | 1.590 | 0.968 | 1.621 | 0.994 | 1.518 | 0.957 | 1.446 | 0.920 | 1.509 | 0.951 | 1.506 | 0.921 | 1.538 | 0.953 | 1.582 | 0.964 | 1.593 | 0.972 |
| | 48 | 1.354 | 0.893 | 1.559 | 0.946 | 1.567 | 0.963 | 1.680 | 1.009 | 1.775 | 1.052 | 1.610 | 0.995 | 1.467 | 0.933 | 1.646 | 0.999 | 1.583 | 0.957 | 1.652 | 1.008 | 1.696 | 1.011 | 1.710 | 1.020 |
| | 60 | 1.383 | 0.913 | 1.602 | 0.971 | 1.609 | 0.988 | 1.776 | 1.053 | 1.958 | 1.122 | 1.679 | 1.020 | 1.626 | 1.006 | 1.727 | 1.043 | 1.693 | 1.003 | 1.752 | 1.049 | 1.796 | 1.061 | 1.829 | 1.064 |
| | Avg | 1.335 | 0.885 | 1.487 | 0.922 | 1.484 | 0.937 | 1.614 | 0.986 | 1.711 | 1.028 | 1.549 | 0.972 | 1.470 | 0.934 | 1.570 | 0.975 | 1.547 | 0.942 | 1.588 | 0.981 | 1.634 | 0.990 | 1.650 | 0.998 |
| METR-LA | 24 | 0.613 | 0.348 | 0.650 | 0.337 | 0.671 | 0.413 | 0.670 | 0.402 | 0.698 | 0.416 | 0.645 | 0.458 | 0.617 | 0.394 | 0.680 | 0.405 | 0.700 | 0.378 | 0.676 | 0.408 | 0.700 | 0.413 | 0.679 | 0.410 |
| | 36 | 0.748 | 0.398 | 0.800 | 0.388 | 0.841 | 0.480 | 0.824 | 0.471 | 0.874 | 0.490 | 0.785 | 0.533 | 0.781 | 0.457 | 0.841 | 0.471 | 0.874 | 0.448 | 0.852 | 0.477 | 0.867 | 0.480 | 0.845 | 0.484 |
| | 48 | 0.860 | 0.438 | 0.905 | 0.427 | 0.964 | 0.531 | 0.955 | 0.521 | 1.013 | 0.546 | 0.885 | 0.585 | 0.842 | 0.520 | 0.963 | 0.528 | 1.017 | 0.498 | 0.982 | 0.526 | 1.017 | 0.539 | 0.972 | 0.536 |
| | 60 | 0.950 | 0.471 | 0.999 | 0.457 | 1.047 | 0.573 | 1.050 | 0.563 | 1.122 | 0.589 | 0.959 | 0.623 | 0.958 | 0.551 | 1.029 | 0.556 | 1.126 | 0.541 | 1.084 | 0.569 | 1.079 | 0.572 | 1.077 | 0.578 |
| | Avg | 0.793 | 0.414 | 0.838 | 0.402 | 0.881 | 0.499 | 0.875 | 0.489 | 0.927 | 0.510 | 0.819 | 0.550 | 0.799 | 0.480 | 0.878 | 0.490 | 0.929 | 0.466 | 0.898 | 0.495 | 0.916 | 0.501 | 0.893 | 0.502 |
| Website | 24 | 0.155 | 0.286 | 0.186 | 0.306 | 0.229 | 0.335 | 0.273 | 0.357 | 0.431 | 0.469 | 0.315 | 0.393 | 0.231 | 0.349 | 0.240 | 0.345 | 0.325 | 0.370 | 0.216 | 0.335 | 0.181 | 0.305 | 0.245 | 0.350 |
| | 36 | 0.224 | 0.343 | 0.272 | 0.356 | 0.361 | 0.420 | 0.401 | 0.441 | 0.554 | 0.552 | 0.385 | 0.447 | 0.328 | 0.397 | 0.327 | 0.405 | 0.428 | 0.442 | 0.331 | 0.411 | 0.226 | 0.343 | 0.370 | 0.429 |
| | 48 | 0.283 | 0.386 | 0.365 | 0.391 | 0.501 | 0.507 | 0.530 | 0.522 | 0.694 | 0.647 | 0.436 | 0.486 | 0.450 | 0.473 | 0.446 | 0.475 | 0.457 | 0.478 | 0.483 | 0.496 | 0.263 | 0.370 | 0.504 | 0.513 |
| | 60 | 0.267 | 0.374 | 0.486 | 0.481 | 0.571 | 0.562 | 0.630 | 0.592 | 0.736 | 0.673 | 0.468 | 0.510 | 0.525 | 0.517 | 0.561 | 0.549 | 0.596 | 0.566 | 0.556 | 0.547 | 0.323 | 0.410 | 0.587 | 0.565 |
| | Avg | 0.233 | 0.348 | 0.327 | 0.383 | 0.415 | 0.456 | 0.458 | 0.478 | 0.604 | 0.585 | 0.401 | 0.459 | 0.384 | 0.434 | 0.393 | 0.443 | 0.451 | 0.464 | 0.396 | 0.447 | 0.248 | 0.357 | 0.426 | 0.464 |
| SP500 | 24 | 0.150 | 0.270 | 0.155 | 0.271 | 0.159 | 0.288 | 0.181 | 0.317 | 0.193 | 0.334 | 0.189 | 0.330 | 0.172 | 0.305 | 0.175 | 0.308 | 0.156 | 0.276 | 0.181 | 0.315 | 0.180 | 0.309 | 0.164 | 0.298 |
| | 36 | 0.204 | 0.322 | 0.209 | 0.317 | 0.218 | 0.343 | 0.224 | 0.341 | 0.259 | 0.389 | 0.250 | 0.363 | 0.227 | 0.344 | 0.232 | 0.358 | 0.206 | 0.319 | 0.239 | 0.365 | 0.225 | 0.346 | 0.221 | 0.341 |
| | 48 | 0.251 | 0.355 | 0.258 | 0.358 | 0.264 | 0.367 | 0.280 | 0.384 | 0.324 | 0.439 | 0.291 | 0.398 | 0.272 | 0.383 | 0.276 | 0.388 | 0.258 | 0.354 | 0.283 | 0.394 | 0.275 | 0.383 | 0.278 | 0.397 |
| | 60 | 0.295 | 0.386 | 0.305 | 0.387 | 0.322 | 0.416 | 0.332 | 0.416 | 0.391 | 0.486 | 0.377 | 0.475 | 0.319 | 0.413 | 0.325 | 0.423 | 0.303 | 0.385 | 0.341 | 0.438 | 0.322 | 0.418 | 0.321 | 0.409 |
| | Avg | 0.225 | 0.333 | 0.231 | 0.333 | 0.241 | 0.353 | 0.254 | 0.365 | 0.291 | 0.412 | 0.277 | 0.391 | 0.247 | 0.361 | 0.252 | 0.369 | 0.231 | 0.333 | 0.261 | 0.378 | 0.250 | 0.364 | 0.246 | 0.361 |
| NASDAQ | 24 | 0.114 | 0.211 | 0.121 | 0.216 | 0.122 | 0.221 | 0.130 | 0.230 | 0.140 | 0.244 | 0.155 | 0.274 | 0.132 | 0.233 | 0.125 | 0.222 | 0.124 | 0.220 | 0.128 | 0.226 | 0.137 | 0.237 | 0.127 | 0.224 |
| | 36 | 0.155 | 0.253 | 0.163 | 0.261 | 0.183 | 0.279 | 0.175 | 0.273 | 0.184 | 0.284 | 0.196 | 0.306 | 0.177 | 0.278 | 0.174 | 0.271 | 0.167 | 0.266 | 0.170 | 0.268 | 0.184 | 0.280 | 0.174 | 0.269 |
| | 48 | 0.196 | 0.290 | 0.205 | 0.296 | 0.200 | 0.298 | 0.224 | 0.314 | 0.234 | 0.324 | 0.244 | 0.344 | 0.216 | 0.311 | 0.222 | 0.312 | 0.218 | 0.307 | 0.218 | 0.306 | 0.229 | 0.318 | 0.225 | 0.314 |
| | 60 | 0.236 | 0.321 | 0.259 | 0.336 | 0.238 | 0.328 | 0.259 | 0.340 | 0.282 | 0.357 | 0.318 | 0.401 | 0.249 | 0.337 | 0.264 | 0.341 | 0.264 | 0.341 | 0.262 | 0.339 | 0.279 | 0.352 | 0.265 | 0.339 |
| | Avg | 0.175 | 0.269 | 0.187 | 0.277 | 0.186 | 0.281 | 0.197 | 0.289 | 0.210 | 0.302 | 0.228 | 0.331 | 0.193 | 0.290 | 0.196 | 0.286 | 0.193 | 0.284 | 0.194 | 0.285 | 0.207 | 0.297 | 0.198 | 0.286 |

7, ReNF remains highly competitive against recent state-of-the-art models in short-term forecasting tasks. However, the performance advantage is less pronounced than in the LTSF setting, a result that aligns with the design principles of BDO. First, when restricted to the shorter look-back windows typical of short-term tasks, the initial forecasts in the BDO process are inherently less accurate. Consequently, recursively combining these noisier predictions can offset the benefits of the paradigm. Second, the hierarchical curriculum learning of BDO yields diminishing returns on short horizons; the strategy of decomposing a short prediction interval into even smaller sub-forecasts becomes redundant. We provide a more rigorous verification of these dynamics in Sec. E.1, where we conduct ablation studies on the BDO module using the M4 and PEMS datasets.

From another perspective, while our proposed methods are universally applicable, these results also highlight the value of more specialized or refined extensions of our proposals in further enhancing the capabilities of BDO in various scenarios.

*Table 8.* Ablation study on the core components of BDO. 'Concat' denotes recursive concatenation in input space; 'Supervision' denotes block-wise supervision. The best results are highlighted in **bold**.

| Method | | ReNF (Ours) | | w/o Concat | | w/o Supervision | |
|---|---|---|---|---|---|---|---|
| | | MSE | MAE | MSE | MAE | MSE | MAE |
| Electricity | 96 | **0.118** | **0.210** | 0.119 | 0.211 | 0.123 | 0.215 |
| | 192 | **0.138** | **0.229** | 0.138 | 0.229 | 0.145 | 0.236 |
| | 336 | **0.151** | **0.244** | 0.153 | 0.245 | 0.156 | 0.251 |
| | 720 | **0.173** | **0.266** | 0.179 | 0.271 | 0.177 | 0.269 |
| | *Avg.* | **0.145** | **0.237** | 0.147 | 0.239 | 0.150 | 0.243 |
| ETTh1 | 96 | **0.350** | 0.383 | 0.356 | **0.382** | 0.356 | **0.382** |
| | 192 | **0.385** | 0.408 | 0.394 | **0.406** | 0.394 | 0.407 |
| | 336 | **0.405** | **0.427** | 0.420 | 0.427 | 0.420 | 0.425 |
| | 720 | **0.422** | **0.449** | 0.432 | 0.454 | 0.436 | 0.456 |
| | *Avg.* | **0.391** | **0.416** | 0.400 | 0.417 | 0.402 | 0.418 |
| ETTm1 | 96 | **0.270** | **0.325** | 0.273 | 0.327 | 0.275 | 0.327 |
| | 192 | **0.310** | **0.352** | 0.311 | 0.355 | 0.312 | 0.353 |
| | 336 | 0.343 | **0.373** | 0.343 | 0.376 | **0.342** | 0.374 |
| | 720 | 0.401 | **0.406** | **0.398** | 0.409 | 0.413 | 0.410 |
| | *Avg.* | **0.331** | **0.364** | **0.331** | 0.367 | 0.336 | 0.366 |
| Solar | 96 | **0.157** | 0.202 | 0.165 | **0.201** | 0.170 | 0.217 |
| | 192 | **0.174** | **0.210** | 0.185 | 0.215 | 0.195 | 0.226 |
| | 336 | **0.180** | **0.219** | 0.185 | **0.219** | 0.189 | 0.233 |
| | 720 | **0.190** | **0.225** | 0.197 | 0.227 | 0.197 | 0.238 |
| | *Avg.* | **0.176** | **0.214** | 0.183 | 0.216 | 0.188 | 0.229 |

# E. Further Ablations

### E.1. The Two Principles of BDO

We posit that our BDO paradigm derives its efficacy from two distinct and essential mechanisms: (1) block-wise Supervision, which generates explicit sub-forecasts from intermediate representations at each stage; and (2) Recursive Concatenation, which feeds these sub-forecasts back into the input of the subsequent stage, encouraging the network to **implicitly learn a combination functional**. To disentangle their respective contributions across diverse forecasting tasks, we conduct ablation studies on these two factors.

**Multivariate Long-Term Forecasting:** Table 8 presents the ablation results on the Electricity, ETTh1, and ETTm1 datasets. The results indicate that the two components are synergistic; removing either leads to a notable degradation in performance. Overall, we observe that hierarchical supervision plays the dominant role in this setting. This is expected, as it not only facilitates the repeated reuse of label information but also enforces a causal and homogeneous structure on the predictive representations across layers—an effect visualized in Figure 11.

**Short-Term Forecasting:** We extend this analysis to short-term tasks using the multivariate PEMS datasets (PEMS03, PEMS08) and the univariate M4 dataset (Yearly, Quarterly, Monthly). For the M4 dataset, we adapt our strategy due to its specific characteristics (fixed, short horizons, and heterogeneous series). We set the length of each sub-forecast equal to the full forecasting horizon. This disables the short-to-long curriculum aspect of block-wise supervision, but keeps the function of implicit forecast combination. As shown in Figure 9, both components remain effective. However, in stark contrast to the long-term setting, the contribution of block-wise supervision is diminished. This aligns with our discussion in Section D.2: when the horizon is short, intermediate supervision steps may introduce auto-regressive noise and error that outweigh the information gain. To further verify this conjecture, Figure 9 illustrates the variation of performance gain provided by block-wise supervision as the forecasting horizon increases. We compare two settings on the PEMS datasets: (1) a short-term setting (forecasting 12, 24, 48 steps given 96 steps) and (2) a long-term setting (forecasting 96, 192, 336 steps given 512 steps). The results clearly show that as the task switches to long-term forecasting, the benefits of block-wise supervision become significant. This confirms that the full BDO paradigm, including its curriculum supervision, is most potent when the input and output contexts are sufficiently long.

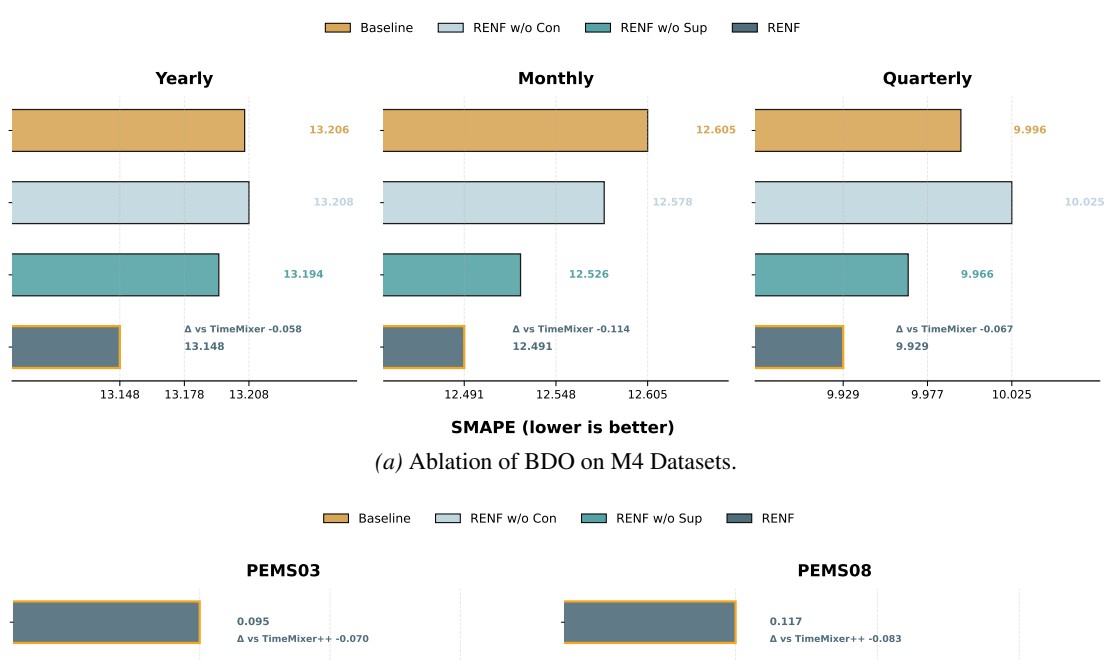

*(a)* Ablation of BDO on M4 Datasets.

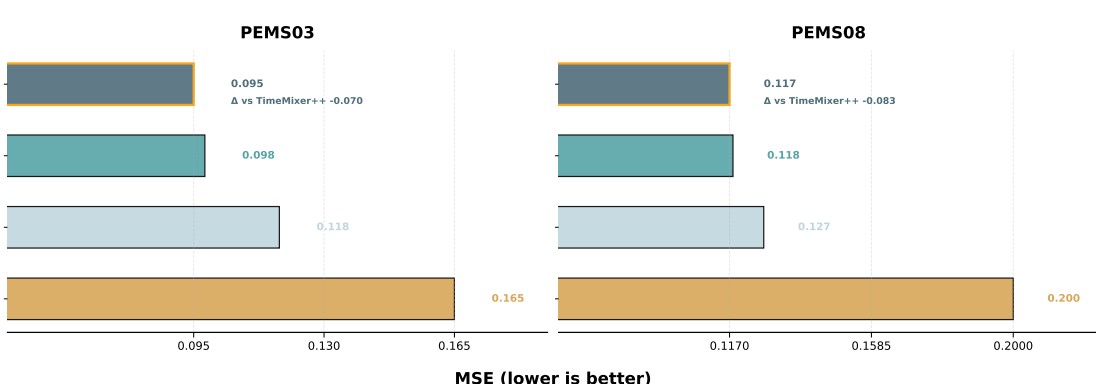

*(b)* Ablation of BDO on PEMS Datasets.

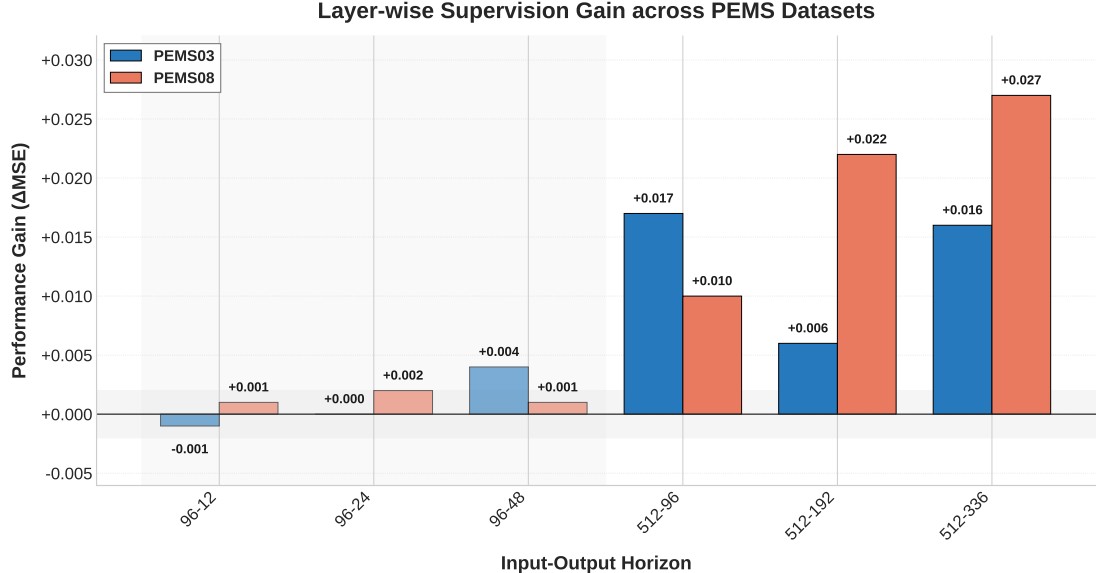

*(c)* Variation of the Layerwise Supervision Gain across PEMS Datasets

*Figure 9.* The ablation of two components of BDO. 'w/o sup' removes the block-wise supervision, and 'w/o con' denotes the ReNF removing the block-wise concatenation (thus losing the function of implicit forecast combination). We select TimeMixer (Wang et al., 2024b) and TimeMixer++ (Wang et al., 2025) as the baseline for M4 and PEMS, respectively. (c) We record the variation of performance gain of block-wise supervision across different forecasting tasks. The advantage of block-wise supervision becomes clearly more significant in the setting of long-term forecasting. All values are obtained by averaging experimental results under three random seeds.

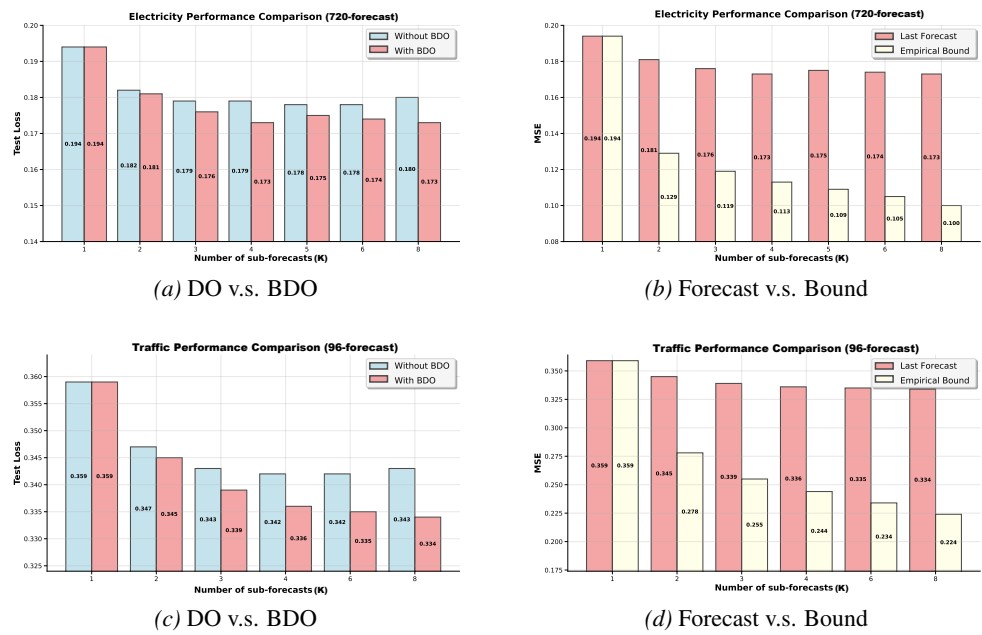

*Figure 10.* Supplementary illustrations of the performance variation with different numbers of sub-forecasts K. (a). Comparison of using DO or BDO with the Electricity dataset. (b).Comparison between the best forecast and the empirical bound of ReNF with the Electricity dataset. (c). Comparison of using DO or BDO with the Traffic dataset. (d).Comparison between the best forecast and the empirical bound of ReNF with the Traffic dataset.

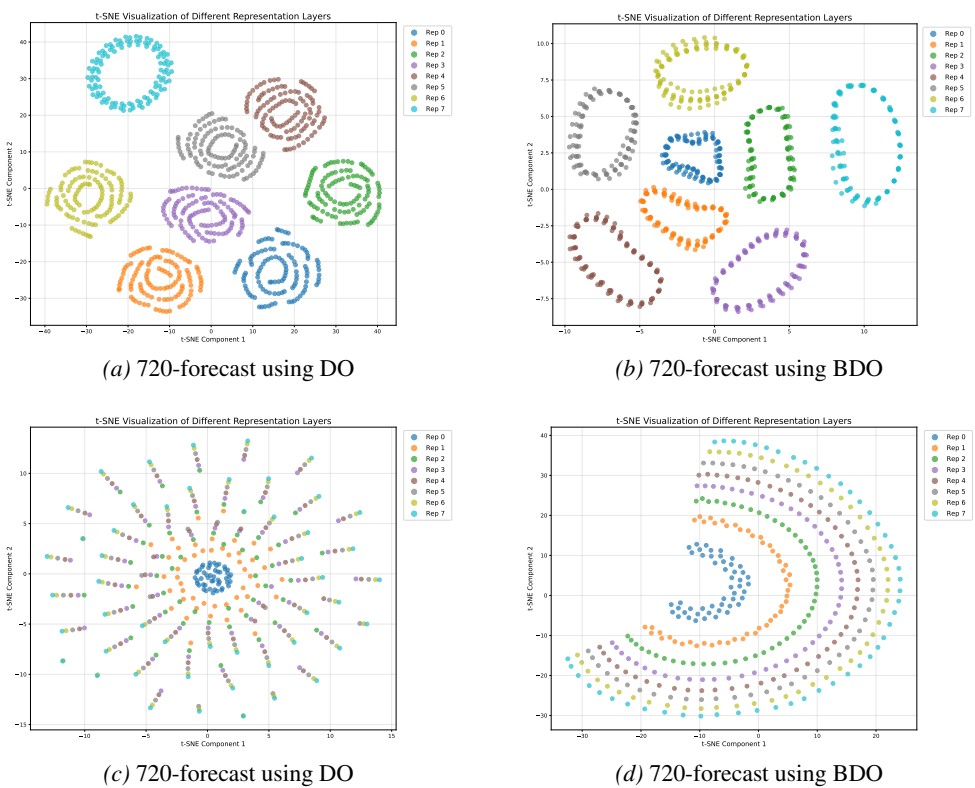

*Figure 11.* Visualizations of the representations of different layers/sub-forecasters. (a). representations of using DO with the ETTh1 dataset. (b). representations of using BDO with the ETTh1 dataset. (c). representations of using DO with the ETTm1 dataset. (d). representations of using BDO with the ETTm1 dataset. This indicates that BDO tends to form predictive representations that are homogeneous and hierarchical.

## E.2. Preference of Deep Representations.

In Sec. 2.3, we introduce two variants of MLPs with a few differences. The primary distinction is the inclusion of skip-connections between representations in ReNF-$\beta$, which facilitates the learning of deeper, more complex non-linear dynamics. To justify this design choice, the following experiments demonstrate the performance degradation that occurs when a model's complexity is mismatched with the dataset's intrinsic characteristics.

*Table 9.* Impact of model complexity on performance. **Origin**: The optimal ReNF configuration found via search. **Exchange**: Results when the ReNF variant ($\alpha$ or $\beta$) is alternated, showing performance degradation when model complexity does not match data characteristics.

| Dataset | | **ReNF (Origin)** | | ReNF (Exchange) | |
|---|---|---|---|---|---|
| | | MSE | MAE | MSE | MAE |
| ETTh1 | 96 | **0.350** | **0.383** | 0.384 | 0.410 |
| | 192 | **0.385** | **0.408** | 0.407 | 0.428 |
| | 336 | **0.405** | **0.425** | 0.447 | 0.456 |
| | 720 | **0.422** | **0.449** | 0.502 | 0.493 |
| | *Avg.* | **0.391** | **0.416** | 0.435 | 0.447 |
| ETTh2 | 96 | **0.261** | **0.329** | 0.271 | 0.332 |
| | 192 | **0.320** | **0.370** | 0.340 | 0.376 |
| | 336 | **0.346** | **0.394** | 0.381 | 0.410 |
| | 720 | **0.381** | **0.423** | 0.415 | 0.437 |
| | *Avg.* | **0.327** | **0.379** | 0.352 | 0.389 |
| Electricity | 96 | **0.118** | **0.210** | 0.125 | 0.217 |
| | 192 | **0.138** | **0.229** | 0.144 | 0.235 |
| | 336 | **0.151** | **0.244** | 0.160 | 0.252 |
| | 720 | **0.173** | **0.266** | 0.196 | 0.283 |
| | *Avg.* | **0.145** | **0.237** | 0.156 | 0.247 |
| Traffic | 96 | **0.335** | **0.226** | 0.359 | 0.241 |
| | 192 | **0.356** | **0.239** | 0.377 | 0.249 |
| | 336 | **0.366** | **0.246** | 0.389 | 0.259 |
| | 720 | **0.402** | **0.267** | 0.426 | 0.284 |
| | *Avg.* | **0.365** | **0.245** | 0.388 | 0.258 |

Specifically, we apply the ReNF-$\beta$ to the ETTh1 and ETTh2 datasets, and apply the alpha version to the large volume Electricity and Traffic datasets. The results are shown in the Table 9, from which we can deduce that the complex deep representations of ReNF-$\beta$ are clearly detrimental to ETTh datasets. In stark contrast, large-volume datasets like Electricity and Traffic benefit from deeper, more expressive representations. This finding provides a direct explanation, from the perspective of representation depth, for the recurring phenomenon where simpler, parsimonious NFs outperform more complex ones on certain benchmarks. It underscores the critical importance of matching model capacity to the intrinsic characteristics of the time series data, identifying a clear direction for future work in adaptive forecaster design.

## E.3. Pre-Dropout.

*Table 10.* Effects of pre-drop.

| Variants | | ETTh1 | | | | | ETTh2 | | | | | Electricity | | | | |
|---|---|---|---|---|---|---|---|---|---|---|---|---|---|---|---|---|
| Metric | | 96 | 192 | 336 | 720 | Avg. | 96 | 192 | 336 | 720 | Avg. | 96 | 192 | 336 | 720 | Avg. |
| ReNF | MSE | 0.350 | 0.385 | 0.405 | 0.422 | 0.391 | 0.261 | 0.320 | 0.346 | 0.381 | 0.327 | 0.118 | 0.138 | 0.151 | 0.173 | 0.145 |
| | MAE | 0.383 | 0.408 | 0.425 | 0.449 | 0.416 | 0.329 | 0.370 | 0.394 | 0.423 | 0.379 | 0.210 | 0.229 | 0.244 | 0.266 | 0.237 |
| ReNF w/o pre-drop | MSE | 0.352 | 0.385 | 0.406 | 0.428 | 0.393 | 0.260 | 0.321 | 0.348 | 0.383 | 0.328 | 0.119 | 0.138 | 0.150 | 0.175 | 0.146 |
| | MAE | 0.386 | 0.410 | 0.429 | 0.454 | 0.420 | 0.329 | 0.370 | 0.396 | 0.425 | 0.380 | 0.210 | 0.229 | 0.243 | 0.267 | 0.237 |

Our MLP architecture applies dropout to the input data by default as a regularization technique. While its impact can be subtle and dataset-dependent, we perform an ablation study for the sake of completeness. The results of this analysis on the ETTh1, ETTh2, and Electricity datasets are presented in Table 10.

| Variants | ReNF origin | | ReNF w/o EMA | |
|---|---|---|---|---|
| Metric | MSE | MAE | MSE | MAE |
| ETTh1 96 | 0.350 | 0.383 | 0.357 | 0.387 |
| 192 | 0.385 | 0.408 | 0.407 | 0.421 |
| 336 | 0.405 | 0.425 | 0.426 | 0.435 |
| 720 | 0.422 | 0.449 | 0.447 | 0.467 |
| *Avg.* | 0.391 | 0.416 | 0.409 | 0.428 |
| ETTh2 96 | 0.261 | 0.329 | 0.265 | 0.329 |
| 192 | 0.320 | 0.370 | 0.336 | 0.375 |
| 336 | 0.346 | 0.394 | 0.397 | 0.418 |
| 720 | 0.381 | 0.423 | 0.421 | 0.441 |
| *Avg.* | 0.327 | 0.379 | 0.355 | 0.391 |
| ETTm1 96 | 0.270 | 0.325 | 0.303 | 0.347 |
| 192 | 0.310 | 0.352 | 0.341 | 0.371 |
| 336 | 0.343 | 0.373 | 0.362 | 0.386 |
| 720 | 0.401 | 0.406 | 0.425 | 0.420 |
| *Avg.* | 0.331 | 0.364 | 0.358 | 0.381 |
| ETTm2 96 | 0.157 | 0.241 | 0.162 | 0.246 |
| 192 | 0.212 | 0.279 | 0.221 | 0.285 |
| 336 | 0.262 | 0.315 | 0.272 | 0.322 |
| 720 | 0.341 | 0.368 | 0.352 | 0.374 |
| *Avg.* | 0.243 | 0.301 | 0.252 | 0.307 |

| Variants | ReNF origin | | ReNF w/o EMA | |
|---|---|---|---|---|
| Metric | MSE | MAE | MSE | MAE |
| Weather 96 | 0.138 | 0.180 | 0.139 | 0.182 |
| 192 | 0.181 | 0.224 | 0.184 | 0.227 |
| 336 | 0.231 | 0.266 | 0.232 | 0.266 |
| 720 | 0.304 | 0.318 | 0.308 | 0.320 |
| *Avg.* | 0.214 | 0.247 | 0.216 | 0.249 |
| Electricity 96 | 0.118 | 0.210 | 0.124 | 0.218 |
| 192 | 0.138 | 0.229 | 0.145 | 0.239 |
| 336 | 0.151 | 0.244 | 0.156 | 0.252 |
| 720 | 0.173 | 0.266 | 0.182 | 0.277 |
| *Avg.* | 0.145 | 0.237 | 0.152 | 0.247 |
| Traffic 96 | 0.335 | 0.226 | 0.341 | 0.237 |
| 192 | 0.356 | 0.239 | 0.363 | 0.249 |
| 336 | 0.366 | 0.246 | 0.373 | 0.256 |
| 720 | 0.402 | 0.267 | 0.411 | 0.278 |
| *Avg.* | 0.365 | 0.245 | 0.372 | 0.255 |
| Solar 96 | 0.157 | 0.202 | 0.177 | 0.232 |
| 192 | 0.174 | 0.210 | 0.193 | 0.234 |
| 336 | 0.180 | 0.219 | 0.195 | 0.239 |
| 720 | 0.190 | 0.225 | 0.199 | 0.242 |
| *Avg.* | 0.176 | 0.214 | 0.191 | 0.237 |

*Table 11.* Full numerical results on the effect of EMA smoothing.

## E.4. EMA Smoothing

The results in Table 11 confirm that EMA smoothing yields substantial improvements in final forecast accuracy. This empirically validates our hypothesis from Section 2: EMA mitigates the detrimental effects of flawed early stopping, where volatile validation scores lead to the premature saving of suboptimal models. By providing a more reliable and stable training signal, EMA not only enhances performance but also establishes the robust foundation necessary to fairly evaluate our other contributions, such as the BDO paradigm.

## F. Robustness

In Table 12, we present the error bar of ReNF in datasets with relatively small sizes or high instability. It shows that ReNF exhibits high robustness because of its simple structure and inherent variance reduction, which is a favorable characteristic for industrial applications. We first calculate the standard deviation for each individual seed of each horizon and then calculate the mean performance across the 4 horizons, see Table 13 as a detailed example of statistical computation.

*Table 12.* Standard deviation and statistical tests for ReNF and TimeBridge on ETTh1, ETTh2, Weather, and Solar datasets. The results are based on the average performance across four prediction lengths from **five runs** with different random seeds. **Note:** 0.000 **means that the deviation is less than** $1e-3$**.** The confidence interval column refers to the statistical significance of our model's outperformance over the baseline (in ours, TimeBridge). We conducted a paired t-test comparing the seed averages of ReNF against those of TimeBridge.

| Model | ReNF | | TimeBridge | | Confidence |
|---|---|---|---|---|---|
| Dataset | MSE | MAE | MSE | MAE | Interval |
| ETTh1 | $0.391 \pm 0.000$ | $0.417 \pm 0.000$ | $0.399 \pm 0.009$ | $0.424 \pm 0.008$ | 99% |
| ETTh2 | $0.327 \pm 0.000$ | $0.380 \pm 0.000$ | $0.343 \pm 0.018$ | $0.383 \pm 0.014$ | 99% |
| Weather | $0.214 \pm 0.000$ | $0.247 \pm 0.000$ | $0.219 \pm 0.006$ | $0.250 \pm 0.003$ | 99% |
| Solar | $0.177 \pm 0.000$ | $0.215 \pm 0.000$ | $0.182 \pm 0.003$ | $0.219 \pm 0.003$ | 99% |

*Table 13.* Supplementary example of the computation in the above table across **three different random seeds**. The results are reported as MSE / MAE.

| Dataset | Horizon | Seed 2021 | Seed 2022 | Seed 2023 | Mean $\pm$ std |
|---|---|---|---|---|---|
| | 96 | 0.118 / 0.210 | 0.117 / 0.210 | 0.119 / 0.211 | $0.118 \pm 0.001$ / $0.210 \pm 0.001$ |
| | 192 | 0.138 / 0.229 | 0.138 / 0.229 | 0.139 / 0.230 | $0.138 \pm 0.001$ / $0.229 \pm 0.001$ |
| **Electricity (ELC)** | 336 | 0.151 / 0.244 | 0.150 / 0.243 | 0.151 / 0.243 | $0.151 \pm 0.001$ / $0.243 \pm 0.001$ |
| | 720 | 0.173 / 0.266 | 0.172 / 0.267 | 0.173 / 0.267 | $0.173 \pm 0.001$ / $0.267 \pm 0.001$ |
| | **Avg.** | — | — | — | $\mathbf{0.145 \pm 0.001}$ / $\mathbf{0.237 \pm 0.001}$ |
| | 96 | 0.270 / 0.325 | 0.270 / 0.326 | 0.270 / 0.325 | $0.270 \pm 0.000$ / $0.325 \pm 0.001$ |
| | 192 | 0.310 / 0.352 | 0.310 / 0.351 | 0.309 / 0.351 | $0.310 \pm 0.001$ / $0.351 \pm 0.001$ |
| **ETTm1** | 336 | 0.343 / 0.373 | 0.343 / 0.374 | 0.342 / 0.373 | $0.343 \pm 0.001$ / $0.373 \pm 0.001$ |
| | 720 | 0.400 / 0.405 | 0.401 / 0.406 | 0.400 / 0.404 | $0.400 \pm 0.001$ / $0.405 \pm 0.001$ |
| | **Avg.** | — | — | — | $\mathbf{0.331 \pm 0.000}$ / $\mathbf{0.364 \pm 0.001}$ |

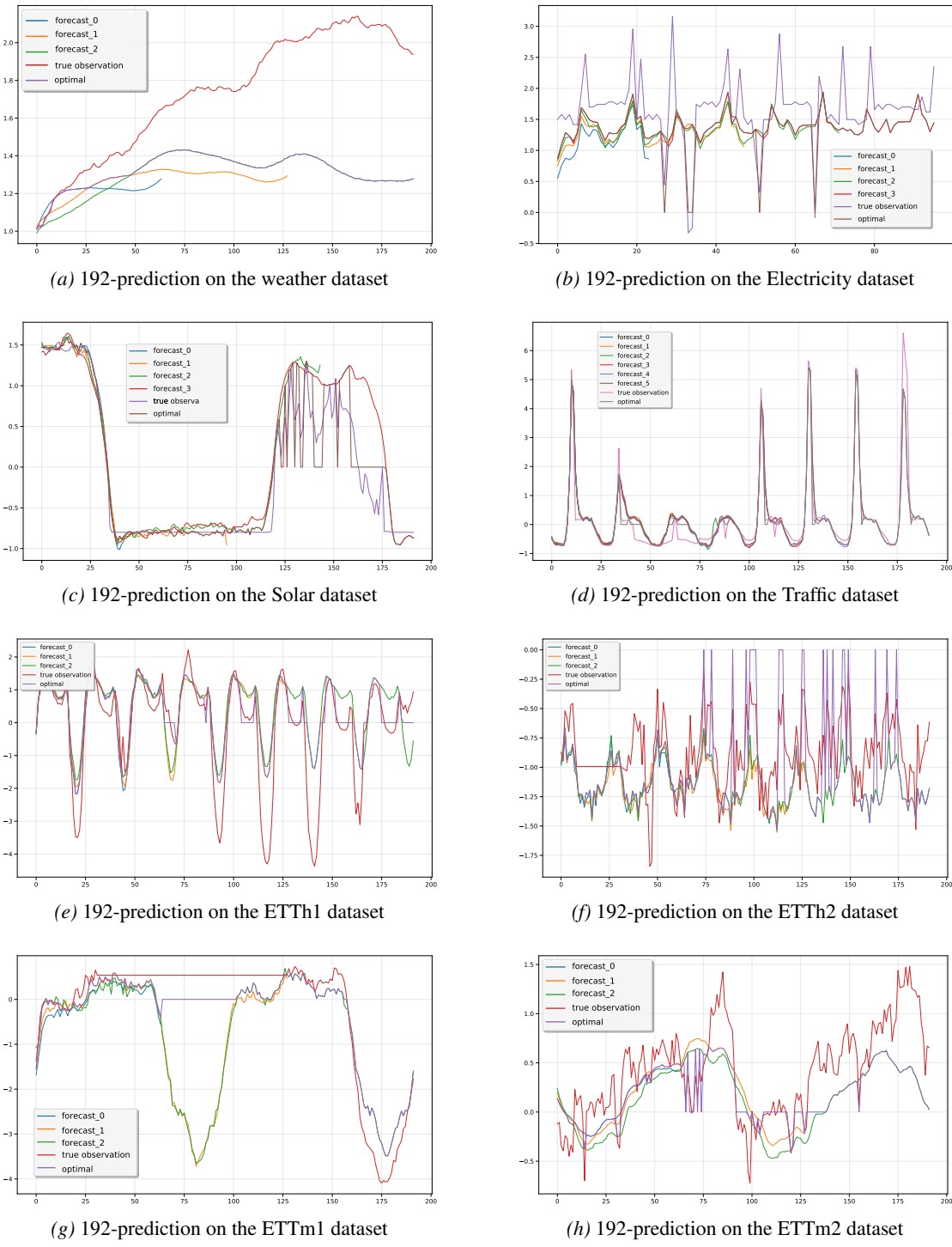

*(a)* 192-prediction on the weather dataset

*(b)* 192-prediction on the Electricity dataset

*(c)* 192-prediction on the Solar dataset

*(d)* 192-prediction on the Traffic dataset

*(e)* 192-prediction on the ETTh1 dataset

*(f)* 192-prediction on the ETTh2 dataset

*(g)* 192-prediction on the ETTm1 dataset

*(h)* 192-prediction on the ETTm2 dataset

*Figure 12.* Visualization of forecasting results of ReNF. The figure shows multiple outputs of ReNF in different layers, along with the result of applying optimal post-combination. The variate of the series in each figure is selected randomly.

