# OpenReview forum: "ReNF: Rethinking the Design of Neural Long-Term Time Series Forecasters"
_ICML.cc/2026/Conference — ICML 2026 regular_

### Official Review · Reviewer_z89X · 2026-03-09

**Soundness:** 2
**Presentation:** 2
**Significance:** 2
**Originality:** 2
**Overall Recommendation:** 4
**Confidence:** 3

**Summary:**

This paper focus on the structural problems of Neural Forecasting in Long-Term Time Series Forecasting (LTSF). The authors argue that current mainstream Direct Output (DO) frameworks lack explicit structural causality, underutilize supervisory signals, lead to prediction variance being ignored, and limit model potential. Based on the Variance Reduction Hypothesis (VRH), this paper proposes a Boosted Direct Output (BDO) prediction paradigm, which achieves implicit prediction combination and variance reduction by recursively generating sub-predictions and introducing hierarchical causality in the structure. In addition, parameter smoothing (e.g., EMA) is used to mitigate the systematic generalization gap between validation-test performance.

The authors evaluate on the mainstream multivariate LTSF dataset and report that their pure MLP architecture can outperform recent SOTA models in this paradigm. The appendix contains derivations of variance decomposition, ablation experiments, and discussion of the method.

**Compliance With Llm Reviewing Policy:**

Affirmed.

**Final Justification:**

See rebuttal acknowledgement.

**Key Questions For Authors:**

1.	Whether theoretical bounds on VRH under the non-independence assumption can be given?

2.	Would replacing the MLP in ReNF with the more advanced backbone lead to greater performance gains?

3.	Does the advantage of EMA come from the longer training time?

4.	Since EMA brings a relatively large performance gain, would combining EMA and TimeBridge/DUET outperform ReNF?

**Limitations:**

The authors explicitly discuss problems of the VRH theory in the appendix.

**Strengths And Weaknesses:**

### Strengths:
1.	The authors rethink and design the forecasting paradigm from the perspective of the lack of structural causality and the neglect of prediction variance in the DO framework. This theoretical motivation is reasonable.
2.	The appendix provides a proof of the hypothesis and clearly distinguishes between the different variance components. Although the theory is preliminary, it has a clear analytical framework.
3.	The experiments are relatively complete, including multiple mainstream LTSF benchmark, key component ablation, empirical upper bound analysis and significance test reports of multiple random seeds.
4.	The structural combination of DO and AR may have potential long-term effects.

### Weaknesses:
1.	The proof of VRH relies on the strong i.i.d. assumption of candidate forecasts, but the sub-forecasts generated by BDO are recursively dependent. There is a theoretical contradiction.

2.	The BDO structure and EMA strategy appear relatively independent in the paper, and the primary and secondary are unclear. In particular, the performance gain of EMA in ablation of Table 11 is significant, and w/o EMA performs worse than TimeBridge or even DUET. However, the comparison between BDO and DO is only presented by partial results (e.g., Figures 7 and 10).

3.	The presentation of the paper needs further improvement:

a)	The model architecture of Figure 3 is not intuitive enough to clearly present the training process and the two core designs of BDO.

b)	The text in some figures is too small to read (e.g., Figures 4, 5, 6, 10, 11).

c)	The legends in Figure 4 seem to be incorrect.

d)	The position of Table 3 is unreasonable and affects reading.

e)	There are some errors in the data annotation in Table 6.

f)	The first letter of the section title should be capitalized.

---

> ### Author Rebuttal · Authors · 2026-03-26
>
> We thank the reviewer for their time and thoughtful feedback. We address each concern below:
>
> ### Regarding Weaknesses:
>
> > W1&Q1: VRH
>
> It is more like a simplification rather than a contradiction. Since ours is not primarily a theoretical paper, we find it sufficient to provide a simplified, loose bound based on the independence assumption to reflect the underlying mechanism. Specifically:
>
> - It provides a **mathematical intuition for our empirical bound experiments in Section 3.3 and aligns well with the results**, which demonstrate that increasing the number of candidate sub-forecasts does indeed increase the accuracy of the ideal combination result.
> - Without this foundational heuristic, our structural analysis comparing the error covariance terms of AR, DO, and BDO (Appendix B.2) would lack a theoretical starting point.
>
> Our intention was indeed to present the Variance Reduction Hypothesis (VRH) as a motivating intuition and heuristic guide, rather than a strict theoretical law.  While a strict theoretical bound under non-independence is an open challenge, we believe that it may be possible to derive a more rigorous bound using concentration formalism without independence. Again, since ours is not primarily a theoretical paper, it would be a promising future work.
>
> > W2 & Q2: BDO
>
> Our original intention of applying EMA was to highlight its effectiveness in stabilizing optimization and mitigating the train/validation inconsistency often caused by non-stationarity in time series, i.e., to ensure a reliable evaluation and early-stopping process. This stability is crucial for our research: to accurately assess the effectiveness of a specific module (like our BDO paradigm), we first need a steady and reliable evaluation to reflect the model's true capability. Without EMA mitigating the valid-test inconsistency, the evaluation noise makes it difficult to validate architectural improvements.
>
> Apart from Figures 7 and 10. We also conducted targeted ablation studies on the M4 and PEMS datasets (Appendix E.1, Figure 9), explicitly demonstrating that BDO's performance advantage may scale with the sequence length.
>
> To make it further, we conducted a preliminary adaptation of BDO to **PatchTST** (the adaptation detail can be seen in our response to reviewer o57a). As shown in the table below, BDO yields overall good improvements for this Transformer backbone, proving it is a general structural enhancement. But the current version of adaptation may not work for all cases; it deserves further exploration with other backbones.
>
> **Please see the second table in our response to the reviewer o57a.**
>
> > W3: About Presentations
>
> Thank you for your suggestions, and we will address them one by one.
>
> ### Regarding Remaining Key Questions:
>
> Q3: Does the advantage of EMA come from the longer training time?
>
> The advantage of EMA does not stem from "longer" training time in a linear sense, but from **effective prolonged training** and more accurate model selection.
>
> As discussed, EMA’s primary role is to smooth the validation signal and mitigate valid-test inconsistency. This prevents **premature early stopping**, allowing the model to continue converging toward a better local minimum rather than being halted by evaluation noise. Thus, the "accuracy boost" is a beneficial byproduct of achieving proper convergence and saving the most stable model state.
>
> Therefore, it may be true that the degree of improvement depends on how much a specific model/dataset suffers from training instability. To demonstrate that this is universal, we applied EMA to PatchTST and TimeBridge. As shown in the table below, these models—which already possess high architectural regularization—exhibit minimal gains from EMA compared to the simpler ReNF.
>
> **Please see the table in our response to the reviewer e5y4.**
>
> Q4:  Would combining EMA and TimeBridge/DUET outperform ReNF?
>
> As indicated by our additional results, it is not guaranteed that applying EMA to complex, advanced models will lead to significant improvements or allow them to outperform ReNF. This observation relates to the broader open question in the field regarding which backbones are most effective for time series forecasting. Also, ReNF's success is not solely due to the "gift" of EMA, but rather the **synergistic combination** of the BDO architecture and the stabilized training process.
>
> However, it can be seen that applying EMA is a generally useful training choice for enhancing both performance and stability across different models.

---

> > ### Author Rebuttal · Reviewer_z89X · 2026-04-01
> >
> > Thank the authors for their response. I will keep my score.

---

> > > ### Author Response · Authors · 2026-04-02
> > >
> > > Thank you for your time. We sincerely appreciate your positive support.

---

### Official Review · Reviewer_e5y4 · 2026-03-13

**Soundness:** 2
**Presentation:** 3
**Significance:** 3
**Originality:** 3
**Overall Recommendation:** 4
**Confidence:** 4

**Summary:**

This paper argues that recent long-term time series forecasting work has focused too much on architecture and not enough on forecasting principles. The authors propose two main ideas: Boosted Direct Output (BDO), which recursively generates progressively longer sub-forecasts instead of predicting the whole horizon at once, and EMA-based parameter smoothing to reduce the validation-test inconsistency during training. These are instantiated in a simple MLP-based model, ReNF, which performs strongly on standard LTSF benchmarks. The paper also introduces a Variance Reduction Hypothesis (VRH) to motivate using multiple sub-forecasts.

**Compliance With Llm Reviewing Policy:**

Affirmed.

**Final Justification:**

I ended up positive on this paper because I think the empirical contribution is real and useful. The main takeaway for me is that forecasting structure matters, not just backbone design, and the paper demonstrates that clearly. BDO is a simple and sensible idea, and it seems to help consistently, especially in the harder settings. The EMA part also looks practically useful.

My main hesitation is still that some of the broader framing goes beyond what is actually shown. In particular, I do not think VRH is established as a real explanation so much as a helpful intuition, and the broader EMA claim should be stated more carefully. That said, the rebuttal addressed these concerns in a reasonable way and made it clear that the authors are willing to soften those claims and improve the limitations discussion.

So overall, I see this as a strong empirical paper with some overstated framing, rather than a paper with weak substance. With the claims better calibrated, I think it is worthy of weak accept.

**Key Questions For Authors:**

In the main text, the VRH-related bound is explicitly stated to rely on independence assumptions and to be only a qualitative guide. Given that later BDO sub-forecasts depend on earlier predicted outputs, how should readers interpret VRH beyond a motivating intuition?

The paper argues that EMA is a necessary protocol for robust performance. Is this meant as a claim specifically about ReNF, or as a broader claim about neural forecasters in LTSF? If it is the latter, I think the wording should be softened or the evidence expanded.

The BDO ablations are encouraging. Are the gains mainly concentrated in the hardest long-horizon settings, or do they remain strong more generally?

**Limitations:**

I do not think the paper discusses limitations adequately. In particular, it should be more explicit that the VRH analysis is heuristic, that the empirical evidence is stronger than the theoretical justification, and that the broader EMA claim is only partially supported by the experiments shown.

**Strengths And Weaknesses:**

Strengths

The empirical result is strong here. A fairly simple MLP doing this well on standard LTSF benchmarks is not trivial, and that part of the paper is convincing. I think the paper makes a useful point that forecasting structure can matter a lot, not just backbone design.

The BDO idea is also reasonably clear. Instead of one-shot direct prediction, the model builds longer forecasts recursively. The ablations suggest this is actually helping and is not just a cosmetic reformulation. In particular, reducing BDO to DO hurts, and using more sub-forecasts generally seems to help, especially at longer horizons.

The EMA part also looks useful in practice. The gains there are strong, and the training seems more stable with it. Even if I am not fully convinced by the broader claims around EMA, I do think this is a practically useful part of the paper.

The paper is also easy enough to follow overall. The method is simple, and that helps.

Weaknesses

My main issue is that the conceptual framing is stronger than what the paper really shows. The paper presents VRH as a central idea and says it is “evaluated and confirmed by experimental results,” but the theory in the main paper already comes with an important qualification: the bound “relies on independence assumptions” and is only a “qualitative guide rather than a strict guarantee.” That matters here because the BDO sub-forecasts are not independent candidates in any clear sense. Later ones depend on earlier predicted outputs. So I do not think the paper really establishes VRH as an explanation of why BDO works and reads more as an intuition.

I also think the EMA claim is too broad as written. The paper says EMA is a “necessary protocol for robust performance” and strongly supports the claim that it addresses valid-test inconsistency in neural forecasting. The experiments do show EMA helps ReNF. But I do not think the broader claim about neural forecasting is supported.

The limitations / broader impacts part is weak. It does not really engage with the fact that the theoretical story is more heuristic than established, and the impact statement is very light.

To me, this works better as a strong empirical paper than as a paper claiming a broader forecasting principle. A recursive direct-output design plus EMA giving a strong simple baseline is already a good contribution. My main issue is that the framing goes beyond what is actually shown. If the VRH and “rethinking principles” claims were toned down, I would be more positive on the paper and closer to weak accept.

---

> ### Author Rebuttal · Authors · 2026-03-26
>
> We thank the reviewer for recognizing the empirical results of our work.
> ### R1: Regarding Weaknesses:
> > W1 & Q1: The role of VRH.
>
> Our intention was indeed to present the Variance Reduction Hypothesis (VRH) as a motivating intuition and heuristic guide, rather than a strict theoretical law. We attempted to convey this in the current manuscript in three ways:
>
> - We did not list VRH as a core contribution in Section 1, as our primary contributions are empirical.
> - In Section 2.1, we introduce VRH as a "core hypothesis that provides the primary motivation," and Remark 2.2 explicitly states it relies on independence assumptions and serves as a "qualitative guide."
> - In Section G (Limitations), we state that VRH is "foundational but preliminary" and serves as a "core intuition."
>
> We include this formalization to show that our intuition is not unfounded. We find it sufficient to provide a simplified, loose-bound based on the independence assumption to reflect the underlying mechanism. This is exactly why we formulate it as a 'hypothesis' to be evaluated, rather than a formal theorem or strict principle:
> - It provides a **mathematical intuition for our empirical bound experiments in Section 3.3 and align well with the results**, which demonstrate that increasing the number of candidate sub-forecasts does indeed increase the accuracy of the ideal combination result.
> - Without this foundational heuristic, our structural analysis comparing the error covariance terms of AR, DO, and BDO (Appendix B.2) would lack a theoretical starting point.
>
> > W2 & Q2: EMA claim is too broad.
>
> We agree that describing EMA as a "necessary protocol" is too strong; we will revise the manuscript to tone down these broad expressions.
> However, we should argue that almost all reviewers focus on the resulting accuracy improvements, which is a highly beneficial byproduct. But it was not our intention to claim EMA as a universal performance booster for all neural forecasters.
> **Our original intention was to highlight its effectiveness in stabilizing optimization and mitigating the inconsistency often caused by non-stationarity in time series, i.e., to ensure a reliable evaluation and early-stopping process.
> This stability is crucial for our research: to accurately assess the effectiveness of a specific module (like our BDO paradigm), we first need a steady and reliable evaluation to reflect the model's true capability.**
>
>
> To the question, here are some empirical results by applying EMA to PatchTST and TimeBridge. As shown below, EMA yields modest improvements or unchanged results for these complex architectures, but the hidden **stabilizing effect** on evaluation is more broadly useful.
> | Model          | Dataset | Setting | w/o EMA (MSE / MAE) |  EMA  |
> | :------------- | :------ | :--------------: | :-----------------: | :---------------: |
> | **PatchTST**   | ETTh1   |     336 - 96     |    0.383 / 0.403    |   0.383 / 0.403   |
> |                | ETTh1   |    336 - 720     |    0.448 / 0.466    |   0.448 / 0.469   |
> |                | ETTh2   |     336 - 96     |    0.275 / 0.338    |   0.275 / 0.338   |
> |                | ETTh2   |    336 - 720     |    0.392 / 0.430    |   0.392 / 0.430   |
> |                | ETTm1   |     336 - 96     |    0.292 / 0.343    | **0.286 / 0.340** |
> |                | ETTm1   |    336 - 720     |    0.417 / 0.423    | **0.410 / 0.417** |
> |                | ETTm2   |     336 - 96     |    0.165 / 0.255    | **0.163 / 0.252** |
> |                | ETTm2   |    336 - 720     |    0.367 / 0.384    | **0.364 / 0.383** |
> | **TimeBridge** | ETTh1   |     720 - 96     |    0.355 / 0.392    |   0.355 / 0.391   |
> |                | ETTh2   |     720 - 96     |    0.271 / 0.331    |   0.271 / 0.331   |
> |                | ETTm1   |     720 - 96     |    0.288 / 0.339    | **0.277 / 0.332** |
> |                | ETTm2   |     720 - 96     |    0.157 / 0.243    | **0.156 / 0.242** |
> |                | Solar   |     720 - 96     |    0.159 / 0.196    | **0.155 / 0.193** |
>
> > W3: The limitation and impact statement part.
>
> We will expand the **Limitations** and will also replace the placeholder **Impact Statement** with a substantive discussion.
> Finally, we will carefully fix any remaining overstatements in the text and be ready to soften the title in the revision.
> ****
> ### R2: Regarding the Remaining Questions
>
> > Q3: Do the gains of BDO remain strong more generally?
>
> As detailed in Appendix D.2 (Table 7), ReNF performs strongly across various short-term benchmarks. However, our targeted ablation studies on the M4 and PEMS datasets (Appendix E.1, Figure 9) explicitly demonstrate that BDO's performance advantage may scale with the sequence length (this may also align with the predictive variance error in VRH).
>
> Specifically, we found that the benefits of BDO's hierarchical curriculum (block-wise supervision) diminish on very short sequences, where intermediate steps can introduce unnecessary auto-regressive noise like AR.

---

> > ### Author Rebuttal · Reviewer_e5y4 · 2026-04-04
> >
> > Thank you for the rebuttal. It addresses my main concern by clarifying that VRH is intended as a heuristic motivation rather than a strict theoretical explanation, and by acknowledging that the EMA claim was too broad and should be softened. These clarifications make the paper’s contribution clearer and better calibrated.
> >
> > I still think the theoretical support for VRH remains limited, and the broader EMA claim is only partially supported. So I continue to view the work primarily as a strong empirical contribution rather than a paper establishing a broader forecasting principle.
> >
> > That said, the empirical results are compelling, and the rebuttal meaningfully improves my confidence in the paper. Assuming the final version tones down the stronger claims and expands the limitations discussion, I update my recommendation to weak accept.

---

> > > ### Author Response · Authors · 2026-04-04
> > >
> > > We really appreciate your support. We will do our best to take your suggestions on board.

---

### Official Review · Reviewer_22aX · 2026-03-13

**Soundness:** 2
**Presentation:** 2
**Significance:** 2
**Originality:** 2
**Overall Recommendation:** 3
**Confidence:** 3

**Summary:**

This paper introduces boosted directed outputs, a method that combines aspects of autoregressive modelling and directed outputs. Suppose we want to generate $kn$ steps in the future. At step $j$ a batch of $n$ predictions are generated. The way this is generated is using the observed data and the sequence of previous predictions. There is a layer that takes in the observed data and produces an output of length $jn$, there is another layer that takes in the sequence of previous predictions (length $(j-1)n$) and gives an output of length $jn$, the two outputs are added together, and this forms the prediction for $jn$ steps. This prediction now gets fed into the next step. This recursive process continues until all $kn$ steps are generated. This maintains autoregressive aspects but also batches of predictions are generated at a time.

In addition, exponential moving average weights is used for stability. Experiments are carried out on a range of multivariate time series forecasting benchmarks.

**Compliance With Llm Reviewing Policy:**

Affirmed.

**Final Justification:**

Overall, the authors have made clarifications and suggested changes that will improve the paper.

However, I am still not entirely satisfied by the responses during rebuttal relating to statistical significance and the calculation of standard deviation to illustrate robustness from random seeds.

Given that VRH raised questions from 3 of the reviewers, it is something that should be handled more carefully in the text, the response is also not very detailed as to how this would be done.

I would prefer to see some of the more substantive changes that the authors promised before acceptance (e.g. VRH, literature/positioning), and therefore have not changed my score.

**Key Questions For Authors:**

For equation (1), the data noise should be time dependent? Or this $\sigma$ does not relate to $\sigma_t$ in the data distribution?

Why do you apply ReNF-$\alpha$ to ETTh1 and ETTh2 but ReNF-$\beta$ to everything else?

It seems to me that the first step/batch of predictions generated can be critical for the subsequent predictions, and it makes sense that these will have greater weight in the loss function. Please can you comment on how sensitive it is to the first prediction, e.g. through seeds?

**Limitations:**

The authors discuss the limitations only in the Appendix, please bring it to the main paper so readers can see these more easily.

**Strengths And Weaknesses:**

__Soundness__

I have some concerns about soundness. Without any sense of standard errors in the results, e.g. Table 1, it is not possible to interpret if these improvements are significant. Similarly, the tuning of EMA parameter seems unstable, it would be useful and important to see how noisy it is across different seeds by adding error bars to the plots.

Another concern is that the authors plot validation and test losses during training. For the scientific process, the test set should not be evaluated on alongside the validation set during training. Only once the model is fixed. Otherwise the test set seems to be treated as a second validation set. The claim "generalization ability of the NF can be substantially enhanced" relies on seeing these test set performance and is not practical to tune in this way.

There are some mistakes Table 1. For electricity Timer-XL gave the second highest MA. There are some numbers which are the same (to 3 decimal places) but have not been coloured in the same way.

Aggregating multiple forecasts into a single one is not always better. If there is a branching point where either you get linear increase or decrease at the same rate afterwards, the average of these paths being a constant line is not useful.

__Originality__

Concatenating the outputs back in to generate the whole prediction sequence again although interesting, is very similar to some existing work in the literature, e.g. those relating to self conditioning or iterative refinement (see e.g. [1] for language modelling application or [2] on RL). Those or similar references should be discussed.

The variance reduction hypothesis through aggregation/additional sampling does not seem that novel. From the central limit theorem, we would expect the variance of the sample mean to scale like $1/\sqrt{N}$, where $N$ is the number of samples.

There is a rich literature and active research on improving the training stability and generalization capabilities, and it would be good to include more references when this is discussed on page 1.

Exponential moving average has been quite embedded and people have looked at some general properties, (e.g. [3] and the references within), and is also used extensively in other areas like RL. It would be good to be more thorough with literature on smoothing (e.g. properties also of older work like Polyak-Ruppert averaging), should this be considered a novelty. Otherwise it should not be a key contribution to bring it to forecasting on longer horizons, it is a training choice, like early stopping.

[1] Madaan et al. (2023) SELF-REFINE:Iterative Refinement with Self-Feedback. NeurIPS.
[2] Chen et al. (2023) Planning with Sequence Models through Iterative Energy Minimization. ICLR
[3] Morales-Brotons et al. (2024) Exponential Moving Average of Weights in Deep Learning: Dynamics and Benefits. TMLR.

__Significance__

The experimental results/evaluation on the common benchmarks and out-competing recent models seem like promising results. The gains are modest, and if soundness concerns can be addressed, then it has the potential to have some impact.

__Presentation__

Although in general this is OK, there are places where the writing can be improved.

There are many unused acronyms in the abstract, while MLP is used but not introduced.

The scale of Figure 7(a) and 7(b) are quite different, and it can be misleading for interpretation.

Some notation are not introduced, I think $K$ only appears in Table 3, it is not defined in the text before we see it on page 7.

Both $n$ and $N$ are used at the end of page 3, and $n$ serves as both an index and number of segments, which is confusing.

The arrows, image, and captions for Figure 3 are unclear. The caption discusses "Emb" but this is not even in the figure, is it the PE denoting positional encoding?

It would be good to be explicit about projection dimensions.

Also, it is philosophically debatable whether "progression of any real-world event is a unique, non-repeatable process", and perhaps not ideal as an opening sentence. This seems to say that there are no truly deterministic events. Ideals of stochasticity, measurement error, chaotic behaviour seem conflated in this opening paragraph.

Small typos:

Full stop instead of comma before hypothesis 2.1

In hypothesis 2.1 it should be "mean $\mu_t$" rather than "mean $\mu$

Section heading: conclusion -> Conclusion

The speech marks are the incorrect types

Page 3 "anti-intuitive" -> "counter-intuitive"

Page 5, fourier -> Fourier

---

> ### Author Rebuttal · Authors · 2026-03-26
>
> We thank the reviewer for their time and thoughtful feedback.
> ### R1: Regarding Soundeness
>
> > About the standard errors and robustness.
>
> We provided error bars across five random seeds in **Appendix F (Table 12)** and below. These results show that ReNF exhibits very high robustness (standard deviation < 1e-3 and shown as 0.0 in the table).
> The EMA rate varies because different systems exhibit varying degrees of instability. For a specific dataset, the rate can be fixed, and performance would remain stable, tuned for better presentation.
>
> > validation and test losses during training...
>
> We fully understand and would like to clarify a misunderstanding regarding our evaluation protocol. As stated in the paper, we "recorded the variation of test loss as evidence," and we would like to explain the logic behind this:
>
> During our initial investigations using a simple MLP, we noticed a consistent instability problem: while the validation loss decreased normally, the final test metrics were often **inconsistent with the validation performance** . This made early stopping fail and reliable hyperparameter searching difficult. Based on this observation, we suspected a validation-test gap during the training process. This motivated us to analyze the underlying causes (Sec 2.5) and adopt EMA as a stabilizer.
>
> We plotted the validation and test loss curves **post-hoc** specifically to provide empirical evidence of the existence of this gap and to demonstrate how effectively EMA mitigates it.  Without these comparative plots, our claims regarding the validation-test gap would remain merely intuitive suspicions rather than evidenced findings.
>
> The method is not naive aggregating. Forecast combination is a classic method in time series literature [1].
>
> [1] Timmermann, Allan. "Forecast combinations." Handbook of economic forecasting 1 (2006): 135-196.
> ### R2: Regarding Originality
> Thank you for pointing out these references. While we agree that iterative refinement is a powerful concept across different DL domains, we clarify that BDO is distinct from these paradigms. BDO focuses on **hierarchical horizon expansion and forecast combination** rather than iteratively improving a fixed-length prediction. Unlike standard iterative refinement, we do not **constrain subsequent steps to be "better" versions of the previous output**; instead, each step expands the forecasting horizon while implicitly combining prior sub-forecasts.
>
> We will include more discussions about these references to better position our approach.
>
> > About the VRH.
>
> Please see our response to w1 of reviewer e5y4.
> ### R3: Regarding Presentations
> Thank you for your careful reading. We apologize for any unclear points and will check them one by one.
> ****
> ​Q1: In Equation (1), $\sigma$ represents the expectation of the noise across the entire horizon, effectively $\sigma = \mathbb{E}[\sigma_t]$. While the underlying data noise $\sigma_t$ is time-dependent, we use an ensemble bound to simplify the qualitative analysis of the variance reduction hypothesis.
>
> ​Q2: This choice stems from our empirical investigation into **representation depth**, as detailed in **Appendix E.2 (Table 9)**. We found that smaller datasets, such as ETTh, do not require deep abstract representations and are prone to overfitting with overly complex architectures.
>
> ​Q3: As ReNF is currently a **deterministic model** without stochastic components, random seeds only affect weight initialization. Below, the model consistently converges to similar predictive patterns regardless of the initial seed (highly robust).
> | Dataset           | Horizon  |   Seed 2021   |   Seed 2022   |   Seed 2023   |           **Mean ± SD**           |
> | :---------------- | :------: | :-----------: | :-----------: | :-----------: | :-------------------------------: |
> | **ETTh1**         |    96    | 0.350 / 0.383 | 0.350 / 0.382 | 0.350 / 0.383 |   0.350 ± 0.000 / 0.383 ± 0.001   |
> |                   |   192    | 0.385 / 0.408 | 0.385 / 0.409 | 0.385 / 0.409 |   0.385 ± 0.000 / 0.409 ± 0.001   |
> |                   |   336    | 0.405 / 0.425 | 0.405 / 0.424 | 0.406 / 0.426 |   0.405 ± 0.001 / 0.425 ± 0.001   |
> |                   |   720    | 0.422 / 0.449 | 0.424 / 0.450 | 0.422 / 0.448 |   0.423 ± 0.001 / 0.449 ± 0.001   |
> | **Total ETTh1**   | **Avg.** |       —       |       —       |       —       | **0.391 ± 0.000 / 0.416 ± 0.000** |
> | **Weather**       |    96    | 0.138 / 0.180 | 0.138 / 0.180 | 0.138 / 0.180 |   0.138 ± 0.000 / 0.180 ± 0.000   |
> |                   |   192    | 0.182 / 0.224 | 0.182 / 0.225 | 0.182 / 0.225 |   0.182 ± 0.000 / 0.225 ± 0.001   |
> |                   |   336    | 0.231 / 0.266 | 0.231 / 0.267 | 0.232 / 0.266 |   0.231 ± 0.001 / 0.266 ± 0.001   |
> |                   |   720    | 0.304 / 0.318 | 0.305 / 0.318 | 0.305 / 0.318 |   0.305 ± 0.001 / 0.318 ± 0.000   |
> | **Total Weather** | **Avg.** |       —       |       —       |       —       | **0.214 ± 0.000 / 0.247 ± 0.000** |

---

> > ### Author Rebuttal · Reviewer_22aX · 2026-04-03
> >
> > Thank you to the authors for the replies, I have some follow-ups.
> >
> > Table 12 and the provided table only gives the standard deviation for a subset of the datasets used in benchmarking the model, and does not give the complete picture. Please can you provide the s.d. for all datasets? Further, please can you also explain what the $99\%$ confidence interval refers to? In addition, please can you explain mathematically how you obtain the s.d. in the table from the rebuttal? Let us take the MAE of __ETTh1__ dataset as an example. If we were to treat it as $12$ numbers, then I believe I get 0.416 as the mean, and 0.024 as the standard deviation. I believe that Table 12 may have a rounding error, hence 0.417.  If we were first to aggregate into 4 means, then find the mean and the s.d. of these 4 numbers, then I believe I get 0.416, and the 0.024 as the standard deviation. If you were taking an average of the standard deviation from each of the horizons, I get 0.0006, which I find misleading to round down to 0.000 rather than up. However, the more critical issue is that it is misleading to average the standard deviation up this way, they do not aggregate linearly, I don't believe the 0.0006 (misleadingly rounded to 0.000) represent meaningful statistical quantity.
> >
> > I am not suggesting forecast combination is not a classic method in time series literature, in my initial review, I was simply pointing out a limitation of aggregation.
> >
> > Please ensure that the literature of EMA is also similarly discussed in the paper.
> >
> > Although the VRH was not listed in contributions, it nevertheless has a prominent place in the abstract "We begin by formulating a Variance Reduction Hypothesis". I would suggest to restructure the paper and change the tone if this is not a primary contribution and the paper in mainly empirical.

---

> > > ### Author Response · Authors · 2026-04-04
> > >
> > > Thank you for your careful reading and the follow-up questions. We answer them one by one as follows:
> > >
> > > > Full picture of std.
> > >
> > > We'd love to include the complete picture of std for all datasets in the revision. We recorded only four in the appendix, as these four, in our experience, often exhibit the most significant instability in the benchmark due to their small volume or non-stationary nature, and are representative for robustness analyses.
> > >
> > > > How do we get the confidence interval and the reported std.
> > >
> > > In fact, we follow the convention of TimeMixer in ICLR 2024 [1] (table.9~11) and TimeBridge in ICML 2025 [2] (table.15).
> > >
> > > The confidence interval column refers to the statistical significance of our model's outperformance over the baseline (in ours, TimeBridge).  We conducted a **paired t-test comparing the seed-averages** of ReNF against those of TimeBridge. The "99%" indicates that the p-value is less than 0.01, and then we reject the null hypothesis with this confidence.
> > >
> > > About the standard deviation, we appreciate your dedicated checking and give our chance to clarify. The table reports the statistics as in the above-referenced paper. We first calculate the mean performance across the 4 horizons *for each individual seed*, and then calculate the standard deviation of those seed-level averages. The statistics in the original paper were obtained in five random seeds. In the rebuttal, we only report three of them due to the limitation.
> > >
> > > **This is because the goal of the table is to show the *seed robustness* of the model**, which is important in this sub-field, as many proposed models' performance is sensitive to the initialization (random seeds). For instance, if you run the TimeBridge in ETTh1 to predict 96 length, the MSE could be 0.355 in seed 2021 and 0.352 in seed 2023 (varies significantly). In comparison, ReNF is robust in this sense due to its variance reduction nature.
> > >
> > > Furthermore, the seed average test may correspond to your question about the sensitivity of the first prediction.
> > >
> > > However, we agree with your opinion and feel that it needs more clarification in the appendix. Thank you.
> > >
> > > The following provides a sample of two relatively larger datasets,
> > > | Dataset         | Horizon  |   Seed 2021   |   Seed 2022   |   Seed 2023   |           **Mean ± SD**           |
> > > | :-------------- | :------: | :-----------: | :-----------: | :-----------: | :-------------------------------: |
> > > | **ELC**         |    96    | 0.118 / 0.210 | 0.117 / 0.210 | 0.119 / 0.211 |   0.118 ± 0.001 / 0.210 ± 0.001   |
> > > |                 |   192    | 0.138 / 0.229 | 0.138 / 0.229 | 0.139 / 0.230 |   0.138 ± 0.001 / 0.229 ± 0.001   |
> > > |                 |   336    | 0.151 / 0.244 | 0.150 / 0.243 | 0.151 / 0.243 |   0.151 ± 0.001 / 0.243 ± 0.001   |
> > > |                 |   720    | 0.173 / 0.266 | 0.172 / 0.267 | 0.173 / 0.267 |   0.173 ± 0.001 / 0.267 ± 0.001   |
> > > | **Total ELC**   | **Avg.** |       —       |       —       |       —       | **0.145 ± 0.001 / 0.237 ± 0.000** |
> > > | **ETTm1**       |    96    | 0.270 / 0.325 | 0.270 / 0.326 | 0.270 / 0.325 |   0.270 ± 0.000 / 0.325 ± 0.001   |
> > > |                 |   192    | 0.310 / 0.352 | 0.310 / 0.351 | 0.309 / 0.351 |   0.310 ± 0.001 / 0.351 ± 0.001   |
> > > |                 |   336    | 0.343 / 0.373 | 0.343 / 0.374 | 0.342 / 0.373 |   0.343 ± 0.001 / 0.373 ± 0.001   |
> > > |                 |   720    | 0.400 / 0.405 | 0.401 / 0.406 | 0.400 / 0.404 |   0.400 ± 0.001 / 0.405 ± 0.001   |
> > > | **Total ETTm1** | **Avg.** |       —       |       —       |       —       | **0.331 ± 0.000 / 0.364 ± 0.001** |
> > > ****
> > > [1] Wang, Shiyu, et al. "Timemixer: Decomposable multiscale mixing for time series forecasting." arXiv preprint arXiv:2405.14616 (2024).
> > >
> > > [2] Liu, Peiyuan, et al. "TimeBridge: Non-Stationarity Matters for Long-term Time Series Forecasting." International Conference on Machine Learning. PMLR, 2025.
> > >
> > > > Forecasting aggregation.
> > >
> > > We apologize for the misunderstanding. It is meaningful, as simple averaging of very different kinds of forecasting may lead to the problem. For ReNF, its implicit combination actually produces a similar trend in a arbitary interval, more like a fine-tuning.
> > >
> > > > Discussion of EMA
> > >
> > > We value this suggestion. We have already realized the thing and have cited two papers in the manuscript, but we'd love to follow your suggestion to give a necessary discussion.
> > >
> > > > VRH
> > >
> > > Thank you for this kind reminder. We would like to change the expression of these phrases. But we have to place the VRH there since it provides a key intuition of the proposal, and is very important to play as a reference of the empirical evaluation in the experiment part, otherwise the latter is sourceless.

---

### Official Review · Reviewer_o57a · 2026-03-17

**Soundness:** 3
**Presentation:** 3
**Significance:** 3
**Originality:** 3
**Overall Recommendation:** 4
**Confidence:** 2

**Summary:**

This paper studies long-term time series forecasting from forecasting principles rather than architectural design. It considers whether recent progress in neural forecasters has overemphasized complex architectures instead of better forecasting paradigms and training protocols. The authors leverage Variance Reduction Hypothesis (VRH) and use to generate and combine multiple forecasts can reduce predictive uncertainty. With this idea, the paper then proposes Boosted Direct Output (BDO), a recursive forecasting paradigm that blends aspects of autoregressive and direct-output forecasting by repeatedly generating longer-horizon forecasts conditioned on prior sub-forecasts. The resulting model, ReNF, is an MLP-based forecaster with stacked blocks, block-wise supervision, and recursive concatenation, together with EMA parameter averaging. Empirically, this MLP-based model matches or outperforms recent LTSF baselines on many multivariate long-term forecasting benchmarks, and it also includes ablations on different components of the model.

**Compliance With Llm Reviewing Policy:**

Affirmed.

**Key Questions For Authors:**

1. Can you better quantify the contribution of BDO versus EMA? EMA helps substantially across datasets , and BDO also improves results. It would help to see a clearer decomposition of gains from (i) EMA only, (ii) BDO only, and (iii) both together against a matched plain MLP baseline.
2. Why are some reported standard deviations near zero? Please include what sources of randomness were varied.
3. Can the method transfer to stronger backbones beyond the MLP? The appendix notes this as future work . Even one experiment plugging BDO into an existing stronger backbone would help establish whether the contribution is truly problem-specific rather than architecture-specific.

**Limitations:**

Yes

**Strengths And Weaknesses:**

The paper addresses a meaningful question for the LTSF literature. The motivation is well framed: the authors argue that architectural complexity alone may not be the main problem, and that training dynamics and forecasting structure deserve more attention . This is a worthwhile perspective, especially in a field where benchmark gains are often tied to increasingly specialized modules.

The empirical evaluation is broad. On the main LTSF benchmarks, ReNF is competitive with or better than strong recent baselines such as TimeBridge, DUET, TimeDistill, Timer-XL, iTransformer, and TimeMixer on most of the reported average metrics.

My main concern is that the experiment settings. The authors tune ReNF over learning rates, EMA rates, layer counts, model variants, and look-back windows, while it is not clear whether all baselines receive comparably extensive retuning under the same protocol. The paper says baseline results were reproduced with official code and “optimal configurations,” but the precise comparability is not fully documented. Since the paper’s main claim is that simple models can beat complex ones when trained properly, it is important to be especially transparent about this.

---

> ### Author Rebuttal · Authors · 2026-03-26
>
> We thank the reviewer for their time and thoughtful feedback. We address each concern below:
> ### R1: Regarding the experimental settings
>
> We fully understand your concern regarding experimental fairness. Our evaluation protocol follows the well-established **TFB benchmark [1]**, which employs a fixed seed (2021) and performs a meticulous hyperparameter search for each model to ensure that every architecture is evaluated at its optimal performance level. For models not yet included in TFB (such as TimeBridge and TimeXer), we tuned their results using their official codebases under their recommended optimal configurations.
>
> Regarding ReNF, we tuned a specific set of parameters to demonstrate its best performance. According to the authors' experience, the results here may not be simply obtained through parameter tuning. This fact is also noted by Reviewer e5y4, "The empirical result is strong here. A fairly simple MLP doing this well on standard LTSF benchmarks is not trivial..."
>
> [1] Qiu, Xiangfei, et al. "TFB: Towards Comprehensive and Fair Benchmarking of Time Series Forecasting Methods." Proceedings of the VLDB Endowment 17.9 (2024): 2363-2377.
> ****
> ### R2: Key Questions
> > Q1:   Quantify the contribution of BDO versus EMA.
>
> The main effect of EMA is to stabilize the evaluation and help reveal the relatively real performance of the model; without this, some improvements would be hidden due to the inconsistency. That is to say, BDO alone would be beneficial, but sometimes its effect cannot be revealed from the test metric (for example, ETTm1 below) if the wrong model is saved, where the EMA plays its key role as a magnifying glass, especially for ReNF containing multiple forecasters.
> | Dataset         | Horizon | Original (Base MLP) |   EMA only    |   BDO only    | **Full ReNF (BDO+EMA)** |
> | :-------------- | :-----: | :-----------------: | :-----------: | :-----------: | :---------------------: |
> | **ETTh1**       |   96    |    0.361 / 0.384    | 0.356 / 0.383 | 0.356 / 0.386 |    **0.350 / 0.383**    |
> |                 |   720   |    0.447 / 0.466    | 0.434 / 0.455 | 0.436 / 0.459 |    **0.422 / 0.449**    |
> | **Weather**     |   96    |    0.143 / 0.186    | 0.143 / 0.186 | 0.138 / 0.182 |    **0.138 / 0.180**    |
> |                 |   720   |    0.311 / 0.323    | 0.311 / 0.324 | 0.308 / 0.320 |    **0.304 / 0.318**    |
> | **ETTm1**       |   96    |    0.299 / 0.347    | 0.271 / 0.326 | 0.295 / 0.340 |    **0.269 / 0.324**    |
> |                 |   720   |    0.422 / 0.417    | 0.406 / 0.407 | 0.419 / 0.420 |    **0.400 / 0.405**    |
> | **Electricity** |   96    |    0.125 / 0.219    | 0.120 / 0.212 | 0.123 / 0.217 |    **0.118 / 0.210**    |
> |                 |   720   |    0.182 / 0.275    | 0.179 / 0.271 | 0.179 / 0.270 |    **0.173 / 0.266**    |
> > Q2:   About the standard deviations
>
> We run the result on five random seeds and obtain the table. The numbers look tiny because, on the one hand, ReNF exhibits highly robust. On the other hand, it is averaged across four horizons. Please see the table (in reviewer 22aX) of the variation across three random seeds for each forecasting length on ETTh1, which may be clearer evidence that the performance of ReNF is not a "lucky" result.
>
> **Please see the table in our response to the reviewer 22aX.**
>
> > Q3: Can the method transfer to stronger backbones?
>
>  It is entirely possible to apply the ReNF framework (specifically the BDO paradigm) to other backbones. To address this, we evaluated its effectiveness using PatchTST. We performed the necessary structural adjustments to adapt the model to the ReNF architecture: we wrapped the entire PatchTST model (comprising the encoder, a 1-layer middle block, and the decoder) as a single sub-forecaster block, then stacked these blocks to generate forecasts of varying lengths. For simplicity, we utilized the ReNF-alpha version, which does not include skip connections between the representation spaces of different blocks. Preliminary results are as follows (note that we modify on the the official code of PatchTST rather than in the TFB benchmark codebase, which did hyperparameter searches):
> | Dataset     | Horizon | PatchTST (Original) | **PatchTST + BDO** | **Improvement Ratio** (MSE / MAE) |
> | :---------- | :-----: | :-----------------: | :----------------: | :-------------------------------: |
> | **ETTh1**   |   96    |    0.383 / 0.403    | **0.370 / 0.393**  |         **+3.4% / +2.5%**         |
> |             |   720   |    0.458 / 0.474    | **0.440 / 0.459**  |         **+3.9% / +3.2%**         |
> | **ETTh2**   |   96    |    0.275 / 0.337    | **0.273 / 0.335**  |         **+0.7% / +0.6%**         |
> |             |   720   |    0.401 / 0.434    | **0.394 / 0.430**  |         **+1.7% / +0.9%**         |
> | **Weather** |   96    |    0.155 / 0.205    | **0.150 / 0.195**  |         **+3.2% / +4.9%**         |
> |             |   720   |    0.321 / 0.334    |   0.323 / 0.335    |           -0.6% / -0.3%           |

---

### Decision · Program_Chairs · 2026-04-30

**Decision:**

Accept (regular)

**Comment:**

This paper revisits long-term time-series forecasting from the perspective of forecasting structure and training stabilization rather than architectural complexity. The reviews are overall favorable, and the strongest support for the paper is empirical: multiple reviewers found it genuinely informative that improved forecasting structure and stabilization can make even a relatively simple MLP highly competitive on LTSF benchmarks. This is a useful message for the LTSF community, especially in a literature that often emphasizes increasingly specialized architectures. The main reservations concern theoretical and framing maturity, especially that the Variance Reduction Hypothesis (VRH) is better viewed as a heuristic motivation than as a fully established explanation, and that the respective contributions of BDO and EMA should be presented more carefully. The rebuttal addressed these concerns well enough for the three positive reviewers by clarifying the benchmarking protocol, decomposing BDO-only versus EMA-only gains, and softening the stronger theoretical framing. Although one reviewer remained unconvinced, I view this as a strong empirical paper whose main weakness is claim calibration rather than a lack of substance.